# Coherent charge oscillations in a bilayer graphene double quantum dot

K. Hecker[1,2,5] ✉, L. Banszerus [1,2,5], A. Schäpers[1], S. Möller [1,2], A. Peters[1], E. Icking[1,2], K. Watanabe [3], T. Taniguchi [4], C. Volk [1,2] & C. Stampfer [1,2]

The coherent dynamics of a quantum mechanical two-level system passing through an anti-crossing of two energy levels can give rise to Landau-Zener-Stückelberg-Majorana (LZSM) interference. LZSM interference spectroscopy has proven to be a fruitful tool to investigate charge noise and charge decoherence in semiconductor quantum dots (QDs). Recently, bilayer graphene has developed as a promising platform to host highly tunable QDs potentially useful for hosting spin and valley qubits. So far, in this system no coherent oscillations have been observed and little is known about charge noise in this material. Here, we report coherent charge oscillations and $T_2^*$ charge decoherence times in a bilayer graphene double QD. The charge decoherence times are measured independently using LZSM interference and photon assisted tunneling. Both techniques yield $T_2^*$ average values in the range of 400–500 ps. The observation of charge coherence allows to study the origin and spectral distribution of charge noise in future experiments.

The concept of Landau-Zener-Stückelberg-Majorana (LZSM) interference was first introduced to describe atomic collisions[1–5] and has found renewed interest with the advent of artificially designed quantum mechanical two-level systems (TLSs) in a variety of solid-state platforms[6–17]. In particular, LZSM interferometry has been developed into a major work horse to study quantum interference effects in silicon nanowires[6], nitrogen-vacancy centers in diamond[7,8], superconducting qubits[9] and semiconductor quantum dots (QDs), where it allows to coherently control the electron spin[10,11] or the spatial position of an electron in a double quantum dot (DQD)[12–16]. Charge noise limits the charge decoherence time and, e.g., mediated via spin-orbit interaction, the spin decoherence time. Semiconductor QDs have been studied in a wide range of materials, including silicon[18–20], germanium[21] and GaAs[22,23]. More recently, 2D materials, such as bilayer graphene (BLG) and transition metal dichalcogenides have emerged as potentially interesting alternative materials with appealing properties for highly controllable QDs, interesting for hosting spin and valley qubits[24]. BLG offers a gate voltage-tunable band gap[25–27], small spin-orbit interaction and weak hyperfine coupling[24]. Important

achievements in BLG QD research include the implementation of charge detection[28,29], an understanding of spin-valley coupling[30–32], and the measurement of the spin relaxation rate[33,34]. In addition, QDs have also been realized in $WSe_2$ and $MoS_2$ monolayers[35,36], which are of interest due to their substantial intrinsic spin-orbit coupling and potential for light-matter coupling. However, despite these recent experimental advances, no coherent oscillations of either charge, spin or valley states have yet been reported in quantum devices based on 2D materials. A priori, it is not obvious that charge coherence can be observed in van der Waals heterostructures such as BLG encapsulated between hexagonal boron nitride (hBN) crystals. In contrast to QDs in semiconductor heterostructures based on GaAs[37,38] and SiGe[20,39], which are buried tens of nm below the dielectric interface, the electron wave function of a BLG QD extends into the hBN making it susceptible to charge noise present due to disorder at the BLG/hBN interface and to impurities in the hBN. So far, no light has been shed on the role of charge noise in graphene QDs.

Here, we demonstrate coherent charge oscillations in a BLG DQD. In contrast to spin, the charge degree of freedom offers faster

[1]JARA-FIT and 2nd Institute of Physics, RWTH Aachen University, 52074 Aachen, Germany. [2]Peter Grünberg Institute (PGI-9), Forschungszentrum Jülich, 52425 Jülich, Germany. [3]Research Center for Functional Materials, National Institute for Materials Science, 1-1 Namiki, Tsukuba 305-0044, Japan. [4]International Center for Materials Nanoarchitectonics, National Institute for Materials Science, 1-1 Namiki, Tsukuba 305-0044, Japan. [5]These authors contributed equally: K. Hecker, L. Banszerus. ✉e-mail: Katrin.Hecker@rwth-aachen.de

dynamics that can be controlled all-electrically using gate electrodes[40,41]. We operate the DQD in the few electron regime and tune its interdot tunnel coupling to the low GHz regime, which we verify by photon assisted tunneling (PAT) spectroscopy. In a pulsed-gate experiment, we observe LZSM interference oscillations of an excess electron, a characteristic signature of quantum phase coherence. From the PAT experiments, we determine an average ensemble decoherence time $T_2^*$ of around $(416 \pm 110)$ ps, while from the analysis of the LZSM interference oscillations we extract an average decoherence time of around $(483 \pm 24)$ ps. These time scales are en par with those reported for advanced GaAs QDs[41,42] which is a first indicator for low charge noise and an important quality measure for the van-der-Waals interfaces in the BLG-based heterostructure.

## Results

The device used to form a DQD consists of a BLG flake encapsulated between two crystals of hBN placed on a global graphite back gate (BG), with two layers of metallic top gates (i.e., the split and finger gate layer) separated by aluminium oxide. Figure 1a shows a false-color scanning electron microscopy image of the gate structure of the device (see Methods for details)[43]. The BG and split gates (SGs) are used to form a p-type channel connecting source and drain reservoirs. The potential along the channel can be controlled using a set of finger gates (FGs). A DQD is formed by locally overcompensating the potential set by the BG using two adjacent FGs, as schematically depicted in Fig. 1b (see also yellow FGs in Fig. 1a). Figure 1c shows a charge stability diagram of the DQD in the few electron regime. When increasing the FG voltages, especially on the right FG, $V_R$, the current through the DQD increases and the co-tunneling lines become more pronounced. This indicates that the interdot tunnel coupling can be sensitively tuned by the voltages applied to the FGs[44].

The first step towards studying quantum phase coherence in the DQD is to create an effective TLS and to characterize its energy dispersion using PAT spectroscopy. We focus on a single pair of triple

points, where a TLS is formed by a single excess electron that can be located either in the left or the right QD (see Fig. 2a) with the two states $|R\rangle := (N, M+1)$ and $|L\rangle := (N+1, M)$, where $N$ and $M$ is the electron occupation of the left (L) and the right (R) QD, respectively. For large detuning energies, $\varepsilon$, compared to the interdot tunnel coupling energy, $\Delta/2$, the eigenenergies of the TLS are given by $E_L = \varepsilon/2$ and $E_R = -\varepsilon/2$, where $\varepsilon$ is the detuning energy between the two QDs (see white arrow in Fig. 2a and gray dashed lines in Fig. 2b). This approximation becomes invalid for $|\varepsilon| \lesssim \Delta$, where the eigenenergies are given by the more general form $E_\pm = \pm \frac{1}{2}\sqrt{\varepsilon^2 + \Delta^2}$ (see solid lines in Fig. 2b). The state of such a TLS can be represented on the Bloch sphere shown in Fig. 2c, using the states $|R\rangle$ and $|L\rangle$ as a basis.

PAT spectroscopy allows to map out the energy dispersion of the TLS and thereby to determine the tunnel coupling, which together with the detuning fully characterizes the system. This method relies on microwave excitation of electrons across the interdot tunnel barrier, which becomes possible whenever the microwave excitation is in resonance with the energy splitting of the QD states, i.e., $hf = E_+ - E_-$ with the microwave frequency $f$ and Planck's constant $h$[41,45–47]. At a small bias voltage, we detune the energy levels of the QDs such that transport is blocked. An electron in the low energy state (see inset in Fig. 2h) can then only transfer into the higher state if the system absorbs a resonant microwave photon. The excited electron then can tunnel to the reservoir, contributing to a current through the device whose direction depends on the sign of the detuning energy.

Figure 2d shows a charge stability diagram of the triple point in Fig. 2a, recorded at zero bias voltage while applying a microwave excitation of $f = 9$ GHz to the left gate. Two parallel current resonances of opposite sign appear symmetrically around zero detuning. In order to map the energy splitting as a function of detuning, the microwave excitation frequency is varied. In the PAT measurements, also excited states could play a role, if energetically accessible[47]. The absence of these excited states can be explained by the out-of-plane magnetic field of 1.8 T applied to the device, which polarizes the valley states (valley splitting of $\approx 1.5$ meV) and also partly the spin states (spin splitting of $\approx 210$ μeV). The resonances split further with increasing frequency, as shown for $f = 15$ GHz and $f = 25$ GHz in Fig. 2e, f, respectively (see Supplementary Fig. 2 for more data). For a quantitative analysis, we measure the splitting of the PAT resonances along the $V_L$ axis, $\delta V_L$, as a function of the applied microwave frequency, see line cut in Fig. 2g. At $\delta V_L < 0.2$ mV, the PAT resonances begin to overlap, setting a lower bound to the frequency range that can be investigated. The relation of $\delta V_L$ and $f$ is determined by the resonance condition ($hf = E_+ - E_-$) and can be expressed as

$$f(\delta V_L) = \frac{1}{h}\sqrt{(\alpha \delta V_L)^2 + \Delta^2}, \qquad (1)$$

where $\alpha$ is the lever arm that converts the $V_L$ axis to a detuning energy. We fit Eq. (1) to the data with the fit parameters $\alpha$ and $\Delta$ (see red line in Fig. 2h). This yields a tunnel coupling of $\Delta/(2h) = 2.1 \pm 0.2$ GHz and $\alpha = 194.5 \pm 1.1$ μeV/mV, which is in good agreement with the lever arm obtained from DC finite bias spectroscopy measurements (see Supplementary Fig. 1). Additionally, similar measurements were conducted for a set of two different FG voltages ($V_L = 3.475$ V, $V_R = 3.38$ V, and $V_L = 3.555$ V, $V_R = 3.36$ V), which yield $\Delta/2h \approx 1.54$ GHz and $\Delta/2h \approx 7.87$ GHz, respectively. For the following measurements the regime of intermediate tunnel coupling (2.1 GHz) was chosen.

Besides studying the dispersion of the TLS, PAT spectroscopy measurements also probe its coherent properties. From the full width at half maximum, $\gamma$, of the PAT resonance, the ensemble decoherence time $T_{2,PAT}^* = 2h/(\alpha\gamma)$ of the charge degree of freedom can be estimated[41,42,48] The fits in Fig. 2g, yield a $T_{2,PAT}^*$ at the positive (red) and negative (blue) PAT peak of 422 ps and 291 ps, respectively. We assign the timescale extracted in this experiment to the ensemble

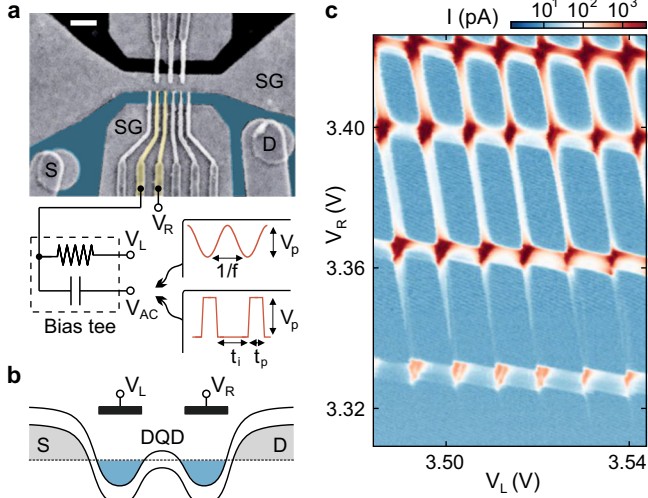

**Fig. 1 | Bilayer graphene double quantum dot. a** False-color scanning electron microscopy image of the DQD device. The scale bar measures 500 nm. The split gates (SG) together with the back gate (not shown) are used to define a narrow conductive channel connecting the source (S) and drain (D) reservoirs (highlighted in blue). The channel is crossed by finger gates (see yellow structures) used to locally tune the band edges to form QDs. Our so-called finger gate left is connected to a bias tee which allow applying DC ($V_L$) and AC voltages ($V_{AC}$). The voltage $V_R$ is applied to the right finger gate. **b** Schematic of the conduction and valence band edge profile of the DQD highlighting the left and right finger gate, where $V_L$ and $V_R$ can be applied. **c** Charge stability diagram of the DQD at low electron occupation measured at a source-drain bias voltage of $V_{SD} = 0.5$ mV.

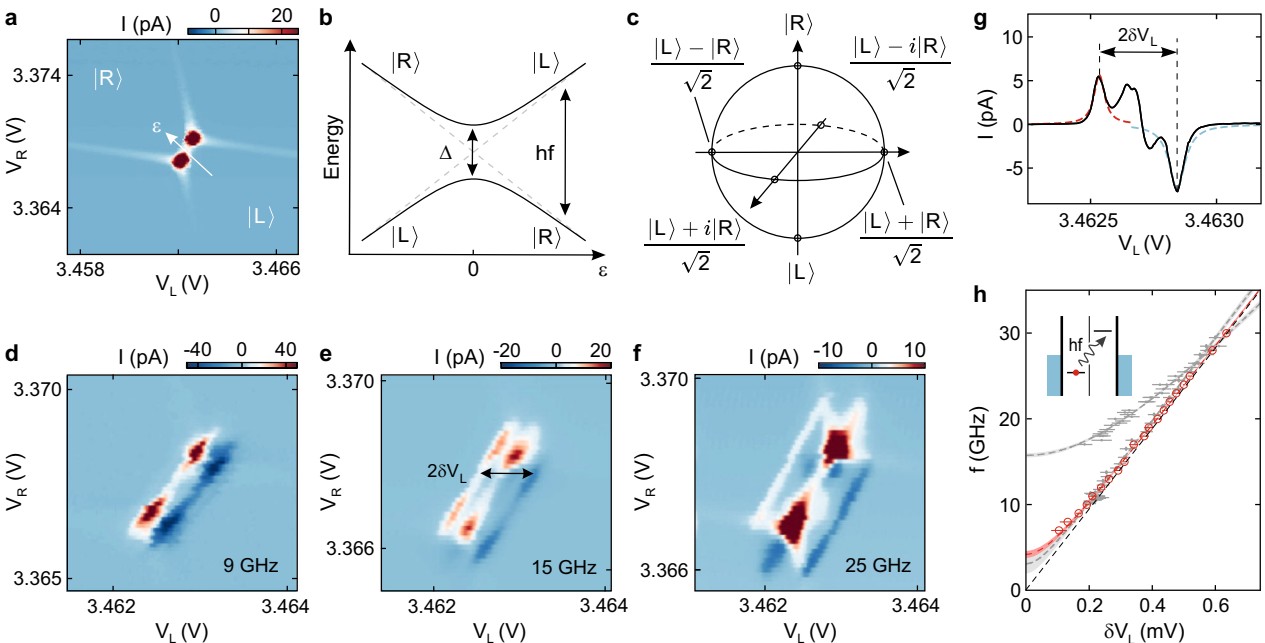

**Fig. 2 | Photon-assisted tunneling. a** Charge stability diagram of a pair of triple points at $V_{SD} = 50\,\mu V$. The charge ground states $|R\rangle$ and $|L\rangle$, corresponding to the position of an excess electron in the DQD (left or right) and the axis of the detuning energy, $\varepsilon$, are indicated. **b** Energy diagram of the TLS. The energies of the uncoupled charge states $|R\rangle$ and $|L\rangle$ are shown by dashed gray lines. For non-zero tunnel coupling, $\Delta \neq 0$, a pair of hybridized eigenstates emerges with eigenenergies represented by the solid black lines, showing a splitting of $\Delta$ at zero detuning. The resonance condition required for PAT is given by $hf = \sqrt{\varepsilon^2 + \Delta^2}$. **c** Bloch sphere representation of the TLS with the charge states $|R\rangle$ and $|L\rangle$ on the poles. **d** Charge stability diagram of the triple point shown in (**a**) while applying a microwave excitation of $f = 9$ GHz and -36 dBm to the left FG. **e**, **f** Measurements as in (**d**) for $f = 15$ GHz and $f = 25$ GHz, respectively. More data sets for different frequencies are presented in

Supplementary Fig. 2. The separation of the peaks, $2\delta V_L$, is measured to investigate the energy splitting of the TLS. **g** Cut through panel **e** along $V_L$. Lorentzians are fitted to the negative (blue) and positive (red) PAT peak. The arrow indicates where $2\delta V_L$ is extracted (see Supplementary Fig. 3 for more data). **h** Resonant excitation frequency as a function of $\delta V_L$ (see **e**). The dashed red curve shows a fit according to $hf = \sqrt{(\alpha\delta V_L)^2 + \Delta^2}$ while the shaded bands mark the $\pm 1\sigma$ confidence interval. The dashed black line is a straight line through the origin with the slope of the lever arm $\alpha = 194.5\,\mu eV/mV$. The gray data points in the background are the results for two distinct sets of FG voltages ($V_L = 3.475$ V, $V_R = 3.38$ V, $\Delta/2h \approx 1.54$ GHz and $V_L = 3.555$ V, $V_R = 3.36$ V, $\Delta/2h \approx 7.87$ GHz). Inset: Schematic representation of the electrochemical potentials in a DQD illustrating the process of PAT.

decoherence time as it results from a current that is integrated over many pulse cycles and can thus be viewed as a time-ensemble average.

In order to obtain a better understanding of the dynamics of the TLS and to gain insight into the charge decoherence time, we perform LZSM interferometry measurements, a powerful technique, where the TLS is driven non-adiabatically through the anti-crossing of the energy levels. To this end, a voltage pulse with a finite rise time, $t_r \approx 140$ ps, is applied to the left finger gate, see Fig. 3a. We initialize (i) the excess electron in the right QD by allowing it to relax into the ground state, $|R\rangle$, at a detuning $\varepsilon_i < 0$ for a time $t_i \approx 3$ ns. To ensure a sufficient initialization but still keep a high signal-to-noise ratio, the initialization time was chosen such that $t_i \gtrsim 1/\Gamma_{comb} \approx 1.6$ ns, where $\Gamma_{comb}$ is the combined tunneling rate through the DQD, estimated from finite bias spectroscopy (see Supplementary Fig. 1). The initial state of the TLS is shown in the level scheme (Fig. 3b (1)) as well as in the energy diagram and Bloch sphere representation (Fig. 3c (1)). After the initialization, the chemical potential of the left QD is shifted by a voltage pulse of nominal amplitude $V_p$, corresponding to an effective detuning pulse of amplitude $A_p$ applied to the sample (see Methods for details). The change in detuning is approximated to occur at a constant rate $v = |\partial\varepsilon/\partial t| \approx 1.6\,\mu eV/ps$. When passing the anti-crossing at zero detuning, the system undergoes a Landau-Zener (LZ) transition from the ground to the excited state with a transition probability given by[49]

$$P_{LZ} = \exp\left(-\frac{\pi}{2}\frac{\Delta^2}{hv}\right), \quad (2)$$

and picks up a relative Stokes phase $\varphi_S$, marked in blue on the equator of the Bloch sphere in Fig. 3c (2). The LZ transition results in a superposition state with weights $1 - P_{LZ}$ and $P_{LZ}$ in the ground and excited state, respectively (Fig. 3b (2), c (2)). The ratio of the weights is experimentally accessible via $v$. In the adiabatic limit ($hv \ll \Delta^2$), the system remains in the ground state, while in the non-adiabatic limit ($hv \gg \Delta^2$), the entire wave function transitions to the excited state. After the first LZ transition, the system is allowed to time-evolve freely at the point of maximum detuning $\varepsilon_i + A_p$ for a time $t_p$ (Fig. 3b (3), c (3)). It accumulates another phase contribution given by[49]

$$\varphi_{ev} = \frac{1}{2h}\int_{t_{LZ_1}}^{t_{LZ_2}} \sqrt{\varepsilon(t)^2 + \Delta^2}\,dt, \quad (3)$$

where $t_{LZ_{1(2)}}$ is the time of the first (second) LZ transition (see gray shaded areas in Fig. 3a, c (3)). While the system is then returned to $\varepsilon_i$ at "rate" $v$, it undergoes a second LZ transition at zero detuning (Fig. 3b (4), c (4)), adding another phase contribution of $\varphi_S$. Both parts of the wave function interfere coherently, such that the final excitation probability is a function of the total relative phase $\varphi = \varphi_{ev} + 2\varphi_S$. Constructive (destructive) interference into the excited (ground) state occurs for $\varphi = 2n\pi$ ($\varphi = (2n+1)\pi$), $n \in \mathbb{N}_0$[13]. At the end of the pulse cycle, a projective readout of the final state is performed. An electron in the excited state $|L\rangle$ can tunnel out of the DQD and contribute to a current that is averaged over many pulse cycles, while an electron in the ground state $|R\rangle$ cannot leave the DQD (Fig. 3b(5)). In the Bloch sphere representation, the readout corresponds to a projection on the $z$ axis (Fig. 3c (5)).

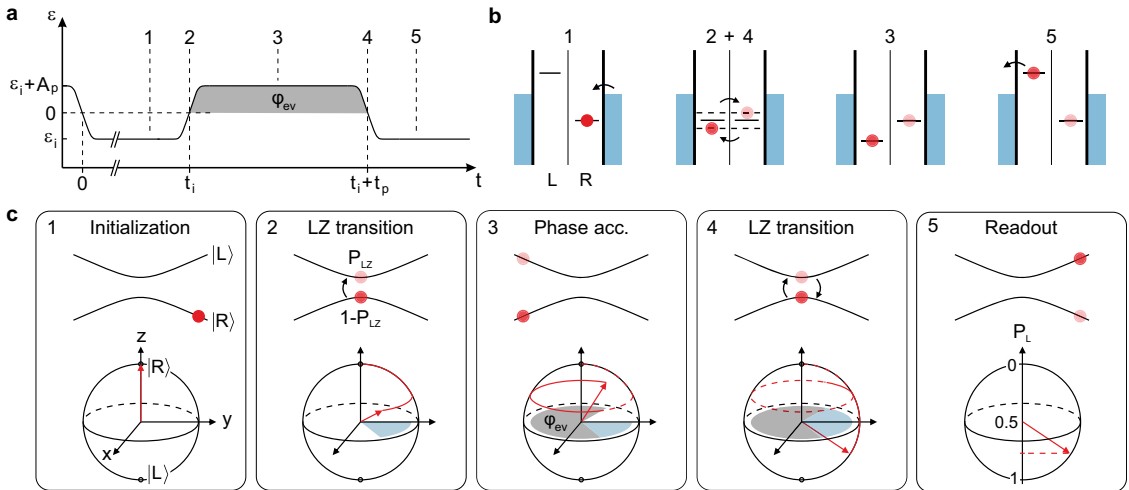

**Fig. 3 | Pulse scheme and state manipulation in a LZSM interference experiment. a** Time-dependent detuning driven by a voltage pulse applied to the left FG characterized by pulse amplitude, $A_p$, and durations $t_p$ and $t_i$. **b** Schematics of the energy levels in the DQD while the pulse scheme in (**a**) is applied. **c** Time evolution of the TLS under the influence of the pulse scheme, shown in the energy diagram and on the Bloch sphere. (1) First, the system is initialized at $\varepsilon_i$ for the duration of $t_i$ (see a). An electron tunnels to the right QD (see **b**) and the system is in the ground state $|R\rangle$, as shown by the red dot and the red arrow. (2) The system is detuned to $\varepsilon_i + A_p$. The process is adiabatic except at the point of $\varepsilon = 0$ (see **a**, **b**), where a LZ transition into the excited state is possible. This creates a superposition state, depicted by two dots in the energy diagram, and a rotation of the Bloch vector about the $x$ axis. In this process, a relative Stokes phase $\varphi_S$ is picked up, represented by the blue shaded area in the equator plane of the Bloch sphere. (3) During $t_p$, both components of the wave function evolve separately in time at a detuning $\varepsilon_i + A_p$, accumulating a relative phase $\varphi_{ev}$ represented by the area shaded in gray. (4) When crossing $\varepsilon = 0$ again (see also panels a,b), a second LZ transition takes place and the initially separated wave functions interfere. (5) In the readout configuration, the detuning is set to $\varepsilon_i$ again (see **a**). An electron in the excited state $|L\rangle$ can tunnel to the drain and contribute to the measured current, while an electron in the ground state would be trapped due to Coulomb blockade (see **b**). By this method the occupation probability of the excited state, $P_L$, is determined.

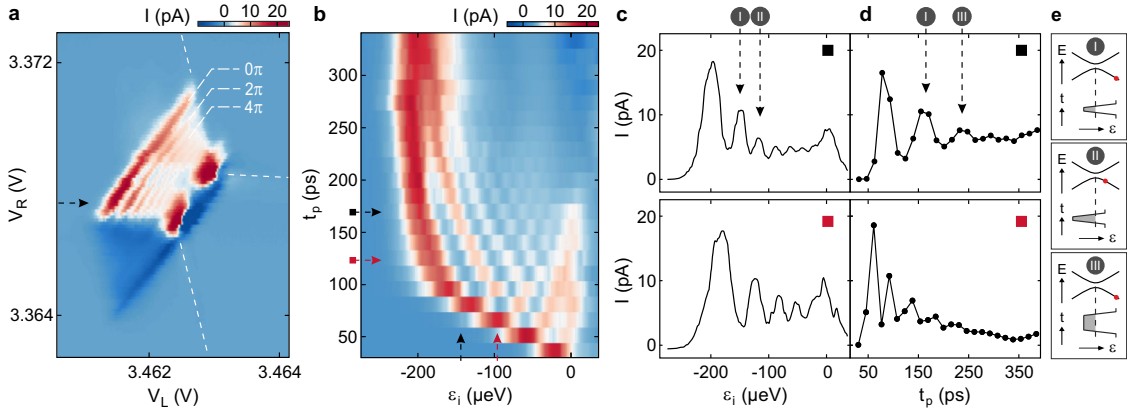

**Fig. 4 | Coherent charge oscillations in the time domain. a** Charge stability diagram of the triple point as in Fig. 2a, while a pulse of duration $t_p = 165$ ps and amplitude $A_p = 228$ μeV is applied to the DQD by the left FG. The black dashed lines highlight the co-tunneling lines. Fringes of increased current indicate coherent charge oscillations. Complementary data sets for different $t_p$ and $A_p$ are presented in Supplementary Fig. 4. **b** Oscillations in the time domain, obtained along the line cut indicated by the black arrow in (**a**). Complementary data sets for different pulse $A_p$ are presented in Supplementary Fig. 5. **c** Cut along the detuning axis as indicated by the horizontal black and red arrows in (**b**). Up to six interference maxima with decreasing intensity can be distinguished. **d** Cut along the $t_p$ axis as indicated by the vertical black and red arrows in (**b**). **e** Schematics illustrating the configurations at the maxima labeled (I)–(III) in (**c**, **d**) indicating the influence of the parameters $\varepsilon_i$ and $t_p$ on the relative phase $\varphi_{ev}$.

Figure 4a shows a charge stability diagram of the transition between $|L\rangle = (N+1,M)$ and $|R\rangle = (N,M+1)$ while applying a voltage pulse to the left FG ($V_L$) as described above. Parallel to the zero detuning line of the triple point, a series of additional lines of increased current can be observed, that are constrained by the co-tunneling lines and the pulse amplitude. This is a clear signature of coherent charge oscillations of one excess electron, where the fringes indicate constructive interference. The first and strongest fringe corresponds to the situation where the pulse just reaches zero detuning, i.e., $\varepsilon_i + A_p = 0$ during the time span $t_p$, and the electron can first tunnel over into the

other QD. At every following line, the relative phase has increased by another $2\pi$. The region of negative current in Fig. 4a can be attributed to charge pumping occuring outside of the gate configurations where the measurement scheme presented in Fig. 3a operates. We find the data in agreement with the adiabatic-impulse approximation[49] as no interference signature is visible before reaching the first fringe.

The loss of charge coherence over time can be explored by performing LZSM interferometry in the time domain, i.e., as a function of the pulse duration $t_p$ and detuning energy during initialization $\varepsilon_i$, as shown in Fig. 4b. An effective pulse amplitude of $A_p \approx 228$ μeV can be

deduced from the position of the first interference fringe at $t_p \approx 200$ ps, which yields a rate of $\nu \approx 1.6$ μeV/ps. Oscillations appear along the detuning and time axes. This observation can be explained by considering how both parameters influence the accumulated phase. Figure 4c plots line cuts along the detuning axis in the time domain (see horizontal dashed lines in Fig. 4b). Up to six interference maxima can be identified before the signal is overlaid by the broad feature at $\varepsilon_i = 0$, which originates from the tunneling of charge carriers through the DQD when the electrochemical potential in the left QD is aligned with the bias during the initialization step. In Fig. 4c, two oscillation maxima are highlighted by (I) and (II) and the acquired phase is indicated by the gray area in the corresponding schematics shown in Fig. 4e. Maximum (I) corresponds to the configuration where a relative phase of $2\pi$ is acquired during one pulse cycle. Following the detuning axis to the next maximum (II), the phase increases by another $2\pi$ as the system is taken further beyond zero detuning, leading to a higher maximum value of the integrand in Eq. (3) (compare gray areas in schematics (I) and (II)). The absence of oscillations on the first fringe ($\varepsilon_i \approx -200$ μeV) in Fig. 4a can be attributed to the distortion of the pulse when transmitted through the setup, as has been studied in GaAs/AlGaAs DQDs[16]. Fig. 4d shows line cuts along the time axis (see vertical dashed lines in Fig. 4b), which show oscillations that are damped due to the loss of quantum phase coherence. However, the resolution along the time axis is not sufficient for a quantitative

decoherence analysis. The expected oscillation period is given by $T = h/\sqrt{\varepsilon^2 + \Delta^2}$[40]. Going from maximum (I) to (III) also adds a phase of $2\pi$, in this case by prolonging $t_p$ and thus expanding the integration bounds in Eq. (3) (compare gray areas in schematics (I) and (III) in Fig. 4e).

Next, we focus on the amplitude domain to quantitatively analyze the loss of phase coherence[9,13,50–52]. The pulse amplitude $A_p$ is another experimental knob to tune the relative phase of the TLS superposition state. Figure 5a shows the current through the device in a pulsed measurement of constant $t_p = 200$ ps, as a function of $\varepsilon_i$ and $A_p$. The data show several parallel interference fringes with decreasing intensity. The first and most prominent fringe, labeled (0), corresponds to the situation sketched in the upper schematic in Fig. 5b, where the pulse takes the system to zero detuning during $t_p$. At the second fringe, labeled (I), $A_p$ has increased such that the pulse crosses zero detuning and a relative phase of $2\pi$ is accumulated, see lower schematic in Fig. 5b. The intensity along a single fringe shows no periodic modulation, which confirms that quantum coherence is only lost between two pulses[6,49]. In the case of multiple coherent LZ transitions, the intensity along a given fringe would additionally be modulated due to the interference of more than two consecutive LZ transitions, whereas only a featureless current would be expected if decoherence occurred between all LZ transitions, i.e., faster than the pulse duration. Hence, the pulse duration ($t_p = 200$ ps) and the initialization time (here:

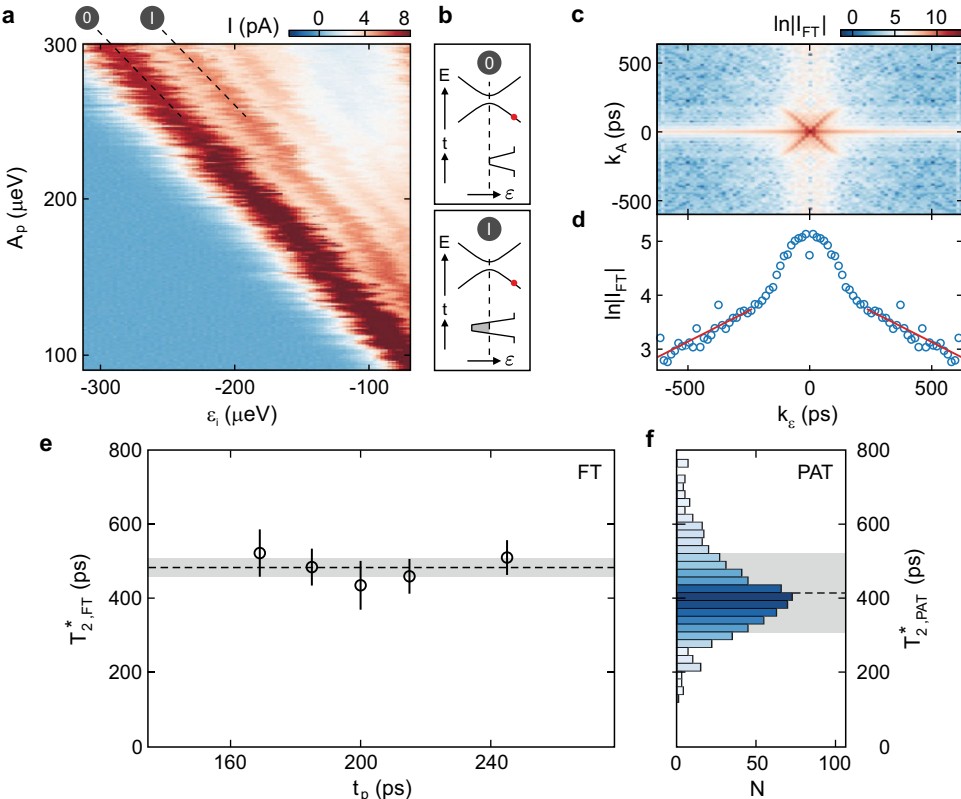

**Fig. 5 | Coherent oscillations in the amplitude domain. a** Current through the device as a function of the initialization detuning $\varepsilon_i$ and pulse amplitude $A_p$, at a constant duration of $t_p = 200$ ps. LZSM interference fringes of two consecutive passages through zero detuning can be seen. Complementary data sets are shown in Supplementary Figs. 6 and 7. **b** Schematics illustrating the pulse and energy diagram on the points (0) and (I) in (**a**). The red dot illustrates the readout/initialization configuration. **c** Fourier transform (FT) of the data in (**a**). The crossing in the center is attributed to the time dependence of the phase[50], while the decreasing background contains information on the decoherence in detuning-space. **d** Line cut along the $k_\varepsilon$ axis averaged over a small area in (**c**). The peak around $k_\varepsilon = 0$ is attributed to low-frequency noise. Red lines are fits to Eq. (5). **e** Decoherence time

$T^*_{2,FT}$ obtained from measurements as in panel a at different pulse widths $t_p$, revealing an average of $T^*_{2,FT} = (483 \pm 24)$ ps (dashed line and gray shaded area). **f** Histogram of $T^*_{2,PAT}$ extracted from the FWHM, $\gamma$, of Lorentzian line shapes fit to the PAT peaks as illustrated in Fig. 2g. $T^*_{2,PAT}$ is calculated according to $T^*_{2,PAT} = 2h/(\alpha\gamma)$[41,42,48]. The histogram contains data points from 18 different data sets measured in a frequency range from $f = 9$ to $30$ GHz. For each frequency, PAT peaks at different $V_R$ have been evaluated. A total of 729 data points for $T^*_{2,PAT}$ are shown in the histogram, where $N$ denotes the number of points in one bin. The average measures $T^*_{2,PAT} = (416 \pm 110)$ ps, highlighted by the dashed black line and the gray shaded area.

$t_i = 3$ ns) give a lower and upper bound for the timescale of decoherence.

In order to quantify the decoherence time, we consider the Fourier transform (FT) of the current signal,

$$I_{FT}(k_\varepsilon, k_A) = \iint \exp(-ik_\varepsilon \varepsilon - ik_A A_p) I(\varepsilon, A_p) d\varepsilon dA_p. \quad (4)$$

The relevant timescale can be obtained from the decay of the Fourier amplitude[13,50,52],

$$\ln|I_{FT}(k_\varepsilon, k_A)| = a - k_\varepsilon/T^*_{2,FT}, \quad (5)$$

where $a$ is an unknown proportionality factor and $T^*_{2,FT}$ is the ensemble decoherence time extracted by this experiment. Figure 5c shows the Fourier transform of the symmetrized current signal (with respect to $A_p$) from Fig. 5a. The prominent cross-like feature contains information on the time between subsequent crossings of zero detuning and the phase accumulated in between[50], but is of no further relevance for the determination of $T^*_{2,FT}$. To obtain $T^*_{2,FT}$, we follow refs. 50–52 and analyze the decay of $\ln(I_{FT})$ as a function of $k_\varepsilon$. Figure 5d shows the average of 100 line cuts through the data in Fig. 5c along $k_\varepsilon$, that do not include the cross-like feature. Similar to other works, the central peak is likely caused by low-frequency noise and therefore not considered in the following[13,51,52]. The red solid lines represent the best fit to Eq. (5) resulting in $T^*_{2,FT} = (435 \pm 66)$ ps. Figure 5e shows $T^*_{2,FT}$ extracted from measurements as in Fig. 5a, c, d for different pulse widths $t_p$. The results yield an average value of $T^*_{2,FT} = (483 \pm 24)$ ps (see dashed line and gray shaded area in Fig. 5e) with an additional systematic uncertainty of $\approx 70$ ps that stems from the uncertainty in determining the lever arm.

## Discussion

For comparison, Fig. 5f shows a histogram of $T^*_{2,PAT}$ extracted from fits to PAT peaks as shown exemplary in Fig. 2g, measured at different excitation frequencies in a range of $f = 9$ to 30 GHz (see also Supplementary Figs. 2 and 3). Evaluating a total of 729 data points, we find a distribution around an average $T^*_{2,PAT} = (416 \pm 110)$ ps (see dashed line and gray shaded area in Fig. 5f), which is slightly smaller than the average $T^*_{2,FT}$ obtained from LZSM interferometry. The significant variation of $T^*_{2,PAT}$, evident from the large error and distribution in Fig. 5f, is likely attributed to slight fluctuations in the power of the applied pulse caused by the frequency-dependent transmission through the setup. The discrepancy between $T^*_{2,PAT}$ and $T^*_{2,FT}$ can be explained by charge fluctuations, which are slower than the coherence time but faster than the integration time of the measurement, and thus may result in a statistical broadening of the PAT resonances. Furthermore, the applied microwave excitation causes a power broadening of the PAT resonances. Both effects lead to an underestimation of $T^*_{2,PAT}$, which therefore only serves as a lower bound. Interestingly, our findings are comparable to the longest decoherence times determined by PAT, $T^*_{2,PAT}$, reported in GaAs (400 ps[41], 250 ps[42]) and are larger than those reported for carbon nanotubes (280 ps[46]), whereas reported $T^*_{2,FT}$ determined by LZSM interference measurements ranges from 100 ps in a Si DQD[52] to 4 ns in a GaAs DQD[13]. When comparing these values with our results, it is important to note that the ensemble decoherence time $T^*_2$ is sensitive to noise contributions from a broad frequency spectrum and depends on the exact measurement configuration. In semiconductor materials, the dominant contributions are intrinsic noise originating from charge fluctuations in the material, as well as charge noise coupling in via the gates[40], where low-frequency noise is assumed to be dominant, typically approximated with an $1/f$ spectral density[42,53–56]. Note that charge noise can affect both $\Delta$ and $\varepsilon$, whereas the latter effect is dominant due to the stronger dependence of $\varepsilon$ on the gate voltages. In first approximation, charge noise couples

in proportionally to the slope of the energy bands, $|dE_\pm/d\varepsilon|$. This leads to an increasing decoherence time when moving closer to the sweet spot $\varepsilon = 0$ where $|dE_\pm/d\varepsilon|$ vanishes[40,42]. In LZSM interferometry, such as in our experiment, the TLS evolves at finite detuning, $\varepsilon \neq 0$, hence $T^*_{2,FT}$ just serves as a lower bound. In particular, the implementation of a charge sensor will allow the direct measurement of coherent charge oscillations (e.g., in a Ramsey experiment) at the sweet spot where the influence of charge noise is further reduced to a minimum promising longer decoherence times[56]. Moreover, in further experiments, including charge echo measurements, the spectral distribution of charge noise in BLG DQDs could be investigated. These techniques have the potential to cancel out the effect of quasi-static noise and increase the charge decoherence time[57,58]. Furthermore, temperature-dependent decoherence measurements allow to quantify electron-phonon coupling, an intrinsic source of decoherence in QDs, as demonstrated in GaAs QDs[40]. Charge coherence experiments can also be extended to DQDs in other 2D materials such as $WSe_2$ and $MoSe_2$[35]. The high sensitivity of this type of experiment can be exploited to characterize material-dependent properties such as defect-induced and interfacial charge noise and electron-phonon coupling, which are expected to be different from BLG.

In summary, we have shown coherent charge oscillations in a BLG DQD, which we discuss in the framework of Landau-Zener-Stückelberg interference. Our findings constitute the first observation of phase coherent oscillations in a graphene QD device and underline this material's potential in the field of quantum technology. We compare the ensemble decoherence times, determined independently from a Fourier analysis of coherent oscillations in the amplitude domain and from PAT spectroscopy. Both methods consistently yield average decoherence times $T^*_2$ in the range of 400–500 ps.

## Methods

The device was fabricated from a BLG flake encapsulated between two hBN crystals of approximately 25 nm (top) and 45 nm (bottom) thicknesses using conventional van-der-Waals stacking techniques. A graphite flake is used as a back gate (BG). The source and drain are etched through the top hBN to contact the BLG using reactive ion etching. The 30 nm thick Cr/Au split gates (SGs) with a lateral separation of 80 nm are deposited on top of the heterostructure. Isolated from the SGs by 15 nm thick atomic layer deposited (ALD) $Al_2O_3$, we fabricate 70 nm wide and 75 nm thick Cr/Au finger gates (FGs) with a pitch of 150 nm. Figure 1a shows a false color scanning electron micrograph of the gate pattern.

In order to perform high frequency gate manipulation, the sample is mounted on a custom-made printed circuit board (PCB). The DC lines are low-pass-filtered (10 nF capacitors to ground). All FGs are connected to on-board bias-tees, allowing for AC and DC control on the same gate (see Fig. 1a). Microwaves are generated by an Agilent E8257D microwave source and pulse sequences are generated by a Keysight M8195A arbitrary waveform generator with a sampling rate of 65 GS/s. The AC signals are attenuated by −10 dB at room temperature and further by −26 dB in the cryostat. The nominal pulse amplitude $V_p$ output by the instruments is converted into an effective amplitude $A_p$ applied to the sample by measuring a charge transition as a function of $\varepsilon$ and $V_p$. This calibration converts the voltage $V_p$ into an energy and takes into account losses due to attenuators, cables and the PCB. For a pulse width $t_p$ exceeding the finite rise time of $t_r \approx 140$ ps, we determine a ratio of $A_p/V_p = 1.50 \pm 0.02$ µeV/mV which is significantly reduced for $t_p < t_r$.

All measurements are performed in a $^3$He/$^4$He dilution refrigerator at a base temperature of around 15 mK and at an electron temperature of around 60 mK using standard DC measurement techniques. The current through the device is amplified and converted into a voltage with a home-built I–V converter at a gain of $10^8$. A p-type channel between source and drain is defined by applying voltages of −3.5 V to

the BG as well as 1.82 V and 1.825 V to the outer and middle SGs, respectively. Positive voltages applied to the left and right FGs, $V_L$ and $V_R$, form a DQD (see Fig. 1b). An out-of-plane magnetic field of 1.8 T has been applied to adjust the tunneling rates.

## Data availability

The data used in this study are available in a Zenodo repository under accession code https://doi.org/10.5281/zenodo.10091584.

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

## Acknowledgements

The authors thank F. Hassler for fruitful discussions, F. Lentz, S. Trellenkamp, M. Otto and D. Neumaier for help with sample fabrication and S. Tautz and M. Ternes for providing access to the AWG Keysight M8195A. This project has received funding from the European Union's Horizon 2020 research and innovation program under grant agreement No. 881603 (Graphene Flagship) and from the European Research Council (ERC) under grant agreement No. 820254, the Deutsche Forschungsgemeinschaft (DFG, German Research Foundation) under Germany's Excellence Strategy - Cluster of Excellence Matter and Light for Quantum Computing (ML4Q) EXC 2004/1 - 390534769, through DFG (STA 1146/11-1), and by the Helmholtz Nano Facility[59]. K.W. and T.T. acknowledge support from JSPS KAKENHI (Grant Numbers 19H05790, 20H00354 and 21H05233).

## Author contributions

L.B., C.V. and C.S. designed the experiment. L.B., K.H., S.M. and E.I. fabricated the device. K.H., L.B., A.S. and A.P. performed the measurements and analyzed the data. K.W. and T.T. synthesized the hBN crystals. C.V. and C.S. supervised the project. K.H., L.B., A.S., C.V. and C.S. wrote the manuscript with contributions from all authors. K.H. and L.B. contributed equally to this work.

## Funding

## Competing interests

The authors declare no competing interests.
