## [Peer Review File · Nature Communications]

The authors reported coherent charge oscillations based on Landau-Zener-Stückelberg interference in a bilayer graphene double quantum dot. They extracted the average ensemble decoherence time in bilayer graphene to be around 400-500 ps from both PAT experiments and LZS interference. Overall, I think the manuscript is well-written. However, I have a few questions and comments that need to be addressed before I can recommend its publication in *Nature Communications*.

1. It is well-known, and is also pointed out by the authors, that graphene is promising for encoding qubits on its spin or valley degrees of freedom. In such sense, what is the novelty for realizing coherent dynamics based on charge degree of freedom in bilayer graphene? Since the estimated coherence time is not significantly longer than those in other material systems, is there any particular interest for demonstrating these charge oscillations in graphene? What would be different if changing graphene to other 2D materials such as MoS₂ or WSe₂? I would like to suggest the authors pay more efforts on discussing these questions to better clarify the novelty of their work.
2. The authors apply an out-of-plane magnetic field of 1.8 T. I am wondering whether spin or/and valley degrees of freedom have played a role in LZS interference.
3. In PAT measurement, the extracted T_2^* decreases rapidly at larger microwave power. How to choose the drive power in the PAT experiment? Are all the data points in Fig. 5f collected at lowest applicable drive power? Is this the origin for large fluctuations found for $T_{2^*,PAT}$?
4. In the Landau-Zener-Stückelberg interference, it is very important to choose an appropriate rate v . What is the value used in the experiment?
5. Related to comment 4, I have a question for Fig. 4b. First, the system is initialized to a particular state, for example $|R\rangle$. Then a pulse is applied to drive the system non-adiabatically to the point of $\epsilon=0$ and wait for a duration time t_p . Finally, the system is detuned for readout. It is expected that charge oscillations can be found when varying t_p , known as Larmor precession (see for example PRL 105, 246804 (2010), Nature 511, 70 (2014), PRL 116, 086801 (2016)). I'm wondering why it is absent in Fig. 4b. It seems that no oscillations are found when the amplitude exactly matches the detuning.
6. What is the repetition rate of the pulse? What is the charge relaxation time T_1 ? The value of t_i is 3 ns in the experiment. Is it sufficient for charge relaxation? The authors should explain and discuss how they choose the parameters of the pulse.
7. The authors directly measure the dot current instead of using a charge sensor. What is the method for the measurement? DC measurement or detecting using a lock-in amplifier? Also, there should be leakage to source/drain via tunneling under microwave driving. Does it bring any influence to the results? I suggest the authors provide

discussions on this.

8. In Fig. 4a, why is there a blue shaded region of negative current?
9. Comparing with Fig. 2d, 2f and Fig. S2, I find that the transition lines locate at different positions in Fig. 2e. Is the figure mislabeled?

Reviewer #2 (Remarks to the Author):

The manuscript “Coherent charge oscillations...” is devoted to the study of driven quantum dots on the base of bilayer graphene. Such structures are perspective given their high tunability and the ability to manipulate electric, spin and valley degrees of freedom. Making use of quantum interferometry, the authors provide the protocol for inducing and detecting coherent charge oscillations and define the charge decoherence times. Thus, the authors demonstrate experimentally the ability of individual control, manipulation, and characterization of the double-quantum-dot devices based on bilayer graphene.

The work is done on the highest experimental level and the paper is very well written. I have only a few comments below. In addition, my report addresses several questions:

- What are the noteworthy results?

The work is devoted to a fine experimental realization of the double quantum dot on the base of a bilayer graphene. This structure is reliably driven, controlled, and measured. Coherent charge oscillations are demonstrated and their analysis allows for obtaining decoherence times, determined from both the Fourier analysis and the photon-assisted tunnelling processes.

- Will the work be of significance to the field and related fields? How does it compare to the established literature?

The work addresses the topical object and uses modern techniques of control, interferometry and spectroscopy. The priority is in observing the multi-photon coherent oscillations in a bilayer graphene double quantum dots for the first time. This seems to be important and promising in the field of quantum engineering and technology.

- Does the work support the conclusions and claims, or is additional evidence needed? Is the methodology sound? Does the work meet the expected standards in your field?

The work is very well written, with all the conclusions and claims well justified. The methodology, including experimental realizations and theoretical analysis, is sound and is on a very high level.

- Is there enough detail provided in the methods for the work to be reproduced?

Experimentally, maybe. Theoretically, yes, there is enough information to reproduce details, including photon-assisted tunnelling (Fig. 2), coherent oscillations (Fig.4), and their Fourier analysis (Fig. 5).

A few particular comments follow.

The work is devoted to a structure based on bilayer graphene. Moreover, it seems that the major ambition of the work is the first demonstration of the coherent oscillations in the quantum structure based on this material. However, I feel that there is not enough information in the paper to convince the reader that coherent control in such structures is so much important and perspective. Very little information is given about the graphene structure in this work and almost no references are given about the *bilayer* graphene, while there are many recent and visible papers on the bilayer graphene, including review articles, e.g.

<https://iopscience.iop.org/article/10.1088/0034-4885/76/5/056503>

<https://doi.org/10.1016/j.physrep.2016.07.003>

The authors refer to the four papers, by Landau, Zener, Stueckelberg, and Majorana, Refs. [1-4]. (Are these references useful for a reader?) However, in the text, the authors use only LZS and LZ abbreviations, why not LZSM? Note that the original works of LZSM were recently discussed here: <https://doi.org/10.1016/j.physrep.2022.10.002>

The caption of Fig. 1(a) says that either the conductive channel or the reservoirs is/are highlighted in blue; but it seems that there is no blue colour in Fig. 1(a).

Also, I guess that in addition to V_L , applied to the left finger, there is V_R applied to the right finger, as we can see in Fig. 1(b). As an optional comment, maybe it is worthwhile to tell something about the right finger, in the caption of Fig. 1(a), after telling about the left one.

Reviewer #3 (Remarks to the Author):

In this manuscript, the authors present an investigation on the microwave spectroscopy of molecular states in a BLG DQDs. A noteworthy contribution of this study is the observation of coherent charge oscillations within the graphene-based system, which has not been reported previously. For the most part, the results and discussions are clear, and the interpretation of the data is warranted.

However, there are several concerns regarding this study that authors need to clarify.

This report focuses solely on a single case (or another, in Fig. S7) of inter-dot tunnel coupling.

The BLG DQDs examined in this study are defined by gate voltages, which allow for control over the inter-dot coupling by manipulating these voltages.

The authors also mention that "This indicates that the interdot tunnel coupling can be sensitively tuned by the voltages applied to the finger gates." In light of this result, it would be valuable to examine the outcomes associated with different tunnel couplings.

The PAT measurement (see Fig. 1h) would clearly estimate the inter-dot tunnel coupling constants.

Pioneering studies of microwave response of DQD in GaAs (T.H. Oosterkamp et al., Nature 395, 873, 1998) have reported the control of the inter-dot tunnel coupling constants by gate voltages.

Furthermore, the period of coherent charge oscillation relies on the inter-dot tunnel coupling constants.

Thus, if the authors can present results demonstrating coherent charge oscillation while controlling the tunnel coupling, it would signify the tremendous tunability of quantum dots within the graphene-based system.

The current manuscript could be read as saying that coherent charge oscillations have been observed in BLG QDs that accidentally possess a moderate inter-dot tunneling coupling. While this outcome is undoubtedly novel, it may not meet the criteria for publication in Nature Communications, which emphasizes research with a broad impact. The authors would need to show that they were able to control BLG DQD with tunnel coupling and coherent charge oscillations tunably. Therefore, I would recommend that the authors consider submitting their current manuscript to a more specialized journal, such as Phys. Rev. Lett.

Other specific comments to be addressed are as follows.

- 1.

In terms of the formation of a quantum two-level system, it would be valuable to elucidate the energy difference of the single-particle level for each individual dot.

This information is essential for a comprehensive understanding of the studied system.

Notably, the presence of 0D-excited states seems to be observed in the lower triple points of Fig. 1(c), and inside the triangle of Fig. S1.

2.

Concerning the PAT experiment, it would be advantageous to demonstrate the microwave power dependence (PAT pattern) and provide an evaluation indicating that the peak heights exhibit behavior consistent with the square of the Bessel function. This evaluation would enhance the understanding of the obtained results.

Reviewer 1:

The authors reported coherent charge oscillations based on Landau-Zener-Stückelberg interference in a bilayer graphene double quantum dot. They extracted the average ensemble decoherence time in bilayer graphene to be around 400-500 ps from both PAT experiments and LZS interference. Overall, I think the manuscript is well-written.

We would like to thank the reviewer for her/his time and interest to examine our manuscript. We are delighted to read, that she/he thinks 'the manuscript is well-written'.

However, I have a few questions and comments that need to be addressed before I can recommend its publication in Nature Communications.

1. It is well-known, and is also pointed out by the authors, that graphene is promising for encoding qubits on its spin or valley degrees of freedom. In such sense, what is the novelty for realizing coherent dynamics based on charge degree of freedom in bilayer graphene? Since the estimated coherence time is not significantly longer than those in other material systems, is there any particular interest for demonstrating these charge oscillations in graphene? What would be different if changing graphene to other 2D materials such as MoS₂ or WSe₂? I would like to suggest the authors pay more efforts on discussing these questions to better clarify the novelty of their work.

We agree with the reviewer that charge coherence experiments are not new. Charge coherence has been studied intensively in conventional semiconductors such as GaAs [1]–[3] and Si [4]. These material systems are well understood by now. However, despite the recent experimental progress in the field of bilayer graphene (BLG) quantum dot (QD) devices, no coherent oscillations have been reported, so far. In fact, it is a priori not at all evident that charge coherence can be observed on similar time scales in van-der-Waals heterostructures of 2D materials – such as the device investigated in our work.

Hence, the novelty of our work lies in demonstrating charge coherence and measuring charge dephasing times in a fundamentally different material system, namely the graphene-based van-der-Waals heterostructure 2D material platform. It is a promising and surprising result that the extracted quantities for the decoherence time are on the same order of magnitude as for conventional semiconductors. This is an important result on the way targeting spin and valley qubits in BLG.

There are significant differences between van-der-Waals heterostructures and conventional two-dimensional electron gases. In a heterostructure of BLG encapsulated between two crystals of hexagonal boron nitride (hBN), the electron wave function will extend into the hBN/BLG interface. Any type of charge disorder present at that interface, e.g. charged impurities, can potentially limit the charge coherence time of the system. In contrast, the two-dimensional electron gas in a GaAs/AlGaAs heterostructure is typically fabricated at a depth of 55-90 nm below the surface [5], [6] while in Si/SiGe heterostructures it is typically buried at 30-60 nm below the surface [7], [8], which makes it less susceptible to disorder at the interface to the dielectric. Techniques like modulation doping are employed to separate dopants from the conventional 2DEGs in order to improve both carrier mobility and charge noise.

Furthermore, in our BLG quantum devices, the thickness of the hBN crystals measures typically about 30 nm, i.e. the gates in closest vicinity to the QDs (back gate and split gates) are only 30 nm away. The small distance together with the large spatial overlap lead to a strong capacitive coupling between these gates and the QDs. This in turn can make the QDs sensitive to gate-induced charge noise limiting the charge decoherence time.

Despite the expected increase in sensitivity to (interface or gate-induced) charge noise, we were able to demonstrate the first coherent control of the charge degree of freedom in any 2D heterostructure. The determined charge decoherence times of 400-500 ps are en par with those reported in GaAs. As outlined above, this finding is not at all obvious and it is also an

indicator for a low density of trap states and hence a quality measure for the van-der-Waals interface.

The high sensitivity of charge coherence measurements to charge noise can be exploited as a method to characterize charge noise in future experiments. The investigation of charge noise is highly relevant also beyond the application as a potential charge qubit: Charge noise can be a source of decoherence and relaxation in any type of qubit with electrical control. In particular, charge noise can couple to the spin degree of freedom via spin-orbit coupling or mediated by a magnetic field gradient induced by a micromagnet. This type of noise has been identified even as the limiting source of decoherence in a Si spin qubit [9]. In addition, calculations show that charge noise is the dominant cause for the relaxation of valley states in BLG QDs in the regime of low magnetic fields (i.e. at low energy splitting) [10]. Hence, charge decoherence measurements are a suitable tool to investigate BLG QD devices and their suitability for application as spin or valley qubits. Furthermore, charge decoherence measurements allow to gain insight in the electron-phonon (el-ph) coupling in BLG QDs. Coupling to acoustic phonons is an intrinsic source of decoherence in semiconductor QDs. This decoherence path typically dominates at zero detuning where the influence of charge noise is minimized. From temperature dependent decoherence measurements, the el-ph coupling can be quantified which has been demonstrated in a GaAs QD device [11].

For what concerns QD devices based on other 2D materials such as WSe₂ and MoSe₂ are an interesting field of research due to their properties distinct from graphene (and BLG), especially their intrinsic spin-orbit coupling. So far, however, these devices suffer from the influence of defects and disorder due to the quality of the flakes or of the interfaces. As described above, we expect that interface disorder in van-der-Waals heterostructures and the defect density in 2D materials will strongly affect charge noise and in turn the decoherence time. Furthermore, the realization of good electric contacts to TMDs, especially at cryogenic temperatures, is still under investigation. In addition, smaller QDs need to be realized accounting for the larger effective mass of the charge carriers in these materials [12], [13].

We have extended the introduction of our manuscript by the following sentences:

"However, despite these recent experimental advances, no coherent oscillations of either charge, spin or valley states have yet been reported in BLG quantum devices. A priori, it is not obvious that charge coherence can be observed in van der Waals heterostructures such as BLG encapsulated between hexagonal boron nitride (hBN) crystals. In contrast to QDs in semiconductor heterostructures based on GaAs [5], [6] and SiGe [7], [8], which are buried tens of nm below the dielectric interface, the electron wave function of a BLG QD extends into the hBN making it susceptible to charge noise present due to disorder at the BLG/hBN interface and to impurities in the hBN. So far, no light has been shed on the role of charge noise in graphene QDs."

"[...] These time scales are en par with those reported for advanced GaAs QDs [2], [3] which is a first indicator for low charge noise and an important quality measure for the van-der-Waals interfaces in the BLG-based heterostructure."

We have added the following sentence to the Discussion section:

"Furthermore, temperature-dependent decoherence measurements allow to quantify electron-phonon coupling, an intrinsic source of decoherence in QDs, as demonstrated in GaAs QDs [11]."

2. The authors apply an out-of-plane magnetic field of 1.8 T. I am wondering whether spin or/and valley degrees of freedom have played a role in LZS interference.

Indeed spin and valley states could potentially play a role in the measurements. Excited states, if energetically accessible, would also show up in the PAT and LZS measurements [14]–[16]. The lack of these observations in our data can be explained by the valley g -factor in BLG QDs which is typically on the order of $g_v \approx 15 - 40$ [17], [18]. At $B_\perp = 1.8$ T, this corresponds to an energy splitting of 1.5–4.3 meV. Concerning the spin, the Zeeman effect shifts the states with opposite spin by $210 \mu\text{eV}$ in energy. Consequently, we can assume full valley and mostly spin polarization within the small detuning range the device is operated in (PAT, Fig. 2: $\varepsilon \leq 140 \mu\text{eV}$, LZS, Fig. 4: $\varepsilon \leq 200 \mu\text{eV}$, Fig. 5: $\varepsilon \leq 300 \mu\text{eV}$).

When applying larger pulse amplitudes, excited states may play a role at some point and would then appear as LZS oscillations in addition to the ground state transition, as can be seen for example in Fig. S6a-c. To better clarify the influence of the magnetic field and the excited state spectrum, we added the following paragraph to the manuscript:

"In the PAT measurements, also excited state could play a role, if energetically accessible [14]. The absence of these excited states can be explained by the out-of-plane magnetic field of 1.8 T applied to the device, which polarizes the valley states (≈ 1.5 meV) and also partly the spin states ($\approx 210 \mu\text{eV}$)."

3. In PAT measurement, the extracted T_2^* decreases rapidly at larger microwave power. How to choose the drive power in the PAT experiment? Are all the data points in Fig. 5f collected at lowest applicable drive power? Is this the origin for large fluctuations found for $T_{2,PAT}^*$?

The applied microwave power in the PAT measurement was chosen sufficiently small to ensure that only the first-order PAT process is visible. Fig. R8a shows a line cut through the triple point as function of the power at room temperature. As an example, we consider the power of 0 dBm as suitable to extract T_2^* here. However, due to frequency-dependent transmission through the setup and the PCB, it was not possible to maintain precisely the same power level for each frequency. Consequently, the data points in Fig. 5f were acquired using the lowest power, where the first-order PAT resonance was observable. We thank the reviewer for noting that these power variations may contribute to fluctuations of T_2^* , resulting in a distributed value of approximately 400 ps, as shown in Fig. 5f. To address this concern, we have revised the text in the manuscript to include the following explanation:

"The significant variation of $T_{2,PAT}^$, evident from the large error and distribution in Fig. 5f, is likely attributed to slight fluctuations in the power of the applied pulse caused by the frequency-dependent transmission through the setup. The discrepancy between $T_{2,PAT}^*$ and $T_{2,FT}^*$ can be explained by charge fluctuations, which are slower than the coherence time but faster than the integration time of the measurement, and thus may result in a statistical broadening of the PAT resonances."*

4. In the Landau-Zener-Stückelberg interference, it is very important to choose an appropriate rate v . What is the value used in the experiment?

We estimate the rate to be $v \approx 1.6 \mu\text{eV/ps}$. The pulse shape has been programmed with a steep flank without intentional rise time. In practice, the rate, v , is given by the finite rise time of the setup. We measured a rise time of about $t_r = 140$ ps at room temperature (see Fig. R1 below), which is in good agreement with the data acquired at low temperature (see Fig. 4b).

This yields a rate of $v \approx 1.6 \mu\text{eV}/\text{ps}$ at a pulse amplitude of $A_p = 230 \mu\text{eV}$. These values have now been included in the manuscript. It now reads:

"To this end, a voltage pulse with a finite rise time, $t_r \approx 140 \text{ ps}$, is applied to the left finger gate, see Fig. 3a." [...] "The change in detuning is approximated to occur at a constant rate $v = |\partial\varepsilon/\partial t| \approx 1.6 \mu\text{eV}/\text{ps}$ " [...] "An effective pulse amplitude of $A_p \approx 228 \mu\text{eV}$ can be deduced from the position of the first interference fringe at $t_p \approx 200 \text{ ps}$, which yields a rate of $v \approx 1.6 \mu\text{eV}/\text{ps}$."

Figure R1: Transmitted signal of a square pulse of amplitude $A = 500 \text{ mV}$ and duration $t_p = 1 \text{ ns}$ through the setup at room temperature.

5. Related to comment 4, I have a question for Fig. 4b. First, the system is initialized to a particular state, for example $|R\rangle$. Then a pulse is applied to drive the system nonadiabatically to the point of $\varepsilon=0$ and wait for a duration time t_p . Finally, the system is detuned for readout. It is expected that charge oscillations can be found when varying t_p , known as Larmor precession (see for example PRL 105, 246804 (2010), Nature 511, 70 (2014), PRL 116, 086801 (2016)). I'm wondering why it is absent in Fig. 4b. It seems that no oscillations are found when the amplitude exactly matches the detuning.

We thank the reviewer for this very relevant question regarding Fig. 4b. The absence of Larmor precession oscillatory behavior on the first fringe, where $\varepsilon = 0$ during t_p , has been observed not only in our experiment but also in GaAs/AlGaAs double quantum dots [19]. In the latter study, the authors found that the 'Rabi-type' oscillations vanish when the pulse shape becomes distorted. In the GaAs/AlGaAs system, the pulse is applied to the drain electrode. When the DQD is isolated from the drain by reducing the QD-lead coupling, the capacitive component leads to pulse distortion and an increased rise time of $t_d = 100 \text{ ps}$. The enhanced rise time due to the distortion was identified as the primary reason for the dominance of the LZS oscillation-like behavior over the Rabi-type oscillations by comparison with simulations.

In our work, we have a rise time of $t_r \approx 140 \text{ ps}$ and a tunnel coupling of $\Delta/2 \approx 8.68 \mu\text{eV}$ which are comparable to the values in the GaAs/AlGaAs device. Therefore, we anticipate observing a similar behavior in our experiment, with the LZS oscillation-like behavior expected to be more prominent due to the influence of the rise time on the pulse shape, similar to the findings in the GaAs/AlGaAs system. We have added the reference to the manuscript:

"The absence of oscillations on the first fringe ($\varepsilon_1 \approx -200 \mu\text{eV}$) in Fig. 4a can be attributed to the distortion of the pulse when transmitted through the setup, as has been studied in GaAs/AlGaAs DQDs [19]."

6. What is the repetition rate of the pulse? What is the charge relaxation time T_1 ? The value of t_i is 3 ns in the experiment. Is it sufficient for charge relaxation? The authors should explain and discuss how they choose the parameters of the pulse.

The repetition rate of $f_{rep} \lesssim 333 \text{ MHz}$ is determined by the pulse duration t_p and initialization, t_i , with t_i dominating as $t_i \gg t_p$. From finite bias measurements we estimated the combined tunnel rate through the DQD to be $\Gamma_{\text{comb}} \gtrsim 630 \text{ MHz} \leq \Gamma_R$, which is faster than the initialization time of $t_i = 3 \text{ ns}$ that we chose. Since the initialization time should be at least longer than the tunneling rate from the right lead, this suggests that t_i is sufficiently long to ensure proper initialization of the quantum states.

For the charge relaxation time, T_1 , we can only give $(2\hbar)/\Delta \approx 0.5 \text{ ns}$ as an order of magnitude. We expect T_1 to be strongly dependent on the inter-dot tunneling rate and slightly dependent on the detuning during the LZS measurement [15]. However, we can only measure the tunnel coupling itself at zero detuning but not depending on the detuning without a charge detector. As a result we can only provide the tunnel coupling as an estimate for T_1 . To clarify the reasoning behind the choice of the pulse parameters, we added the following paragraph to the manuscript:

"To ensure a sufficient initialization but still keep a high signal-to-noise ratio, the initialization time was chosen such that $t_i \gtrsim 1/\Gamma_{\text{comb}} \approx 1.6 \text{ ns}$, where Γ_{comb} is the combined tunneling rate through the DQD, estimated from finite bias spectroscopy (see Fig. S1 in the supplementary)."

7. The authors directly measure the dot current instead of using a charge sensor. What is the method for the measurement? DC measurement or detecting using a lock-in amplifier? Also, there should be leakage to source/drain via tunneling under microwave driving. Does it bring any influence to the results? I suggest the authors provide discussions on this.

We thank the reviewer for raising this point. In all measurements the DC current has been measured using a home-built I-V converter without using a lock-in signal. We extended the information on the measurement principle in the manuscript. It now reads:

"All measurements are performed in a $^3\text{He}/^4\text{He}$ dilution refrigerator at a base temperature of around 15 mK and at an electron temperature of around 60 mK using standard DC measurement techniques. The current through the device is amplified and converted into a voltage with a home-built I-V converter at a gain of 10^8 ."

Concerning the leakage current, there are three sources which are considered in the measurements. The first is co-tunneling appearing along the dashed white lines in Fig. R2a. These second order processes could lead to a current when one QD state is resonant with the leads while the other is not. This effect is also present when a pulse is applied and the co-tunneling lines can further be broadened when heating of the reservoirs resulting from the microwave excitation is appearing. The second source of leakage current is heating of the reservoirs. Depending on the amplitude of the pulse, heating and excitation leads to a smearing of the Fermi edge of the reservoirs. This effect has a similar effect as a virtual bias window and leads to the high currents when the DQD states are aligned with the Fermi energy of the reservoirs at $\varepsilon = 0$ (see Fig. R2). At sufficiently high power, heating can affect the whole DQD system leading to an overall rise of the background leakage current (very faintly seen in Fig. R8). The LZS measurements don't show this amplitude-dependent leakage current away from $\varepsilon = 0$,

as can be seen in Fig. S6, so heating seems to play a minor role in these measurements. Pumping currents are another source of leakage current when applying a microwave pulse to the DQD. They can be seen in Fig. R2a and will be addressed below.

8. In Fig. 4a, why is there a blue shaded region of negative current?

We thank the reviewer for raising this issue and, which is related to pumping currents in the LZS measurements. Fig R2a shows the charge stability diagram, while a square pulse ($f_{\text{rep}} = 316$ MHz, $t_p = 165$ ps, $t_i = 3$ ns, $A_p = 228 \mu\text{eV}$) is applied to the left FG. The average current through the DQD is measured from source to drain. During the dynamic manipulation of the DQD, additional negative currents appear due to charge pumping. The schematics in Fig R2b show the level configuration of the electrochemical potentials at the positions in gate space denoted by the numbers i-iv in Fig. R2a during the initialization part of the pulse, t_i , (blue lines) and during t_p (red lines). The triangular shaped region of negative current in Fig. R2a can be attributed to an inversion of the pulse cycle. As shown in schematics ii and iii, the DQD is initialized in $|L\rangle$ during t_p . The level of the left QD then is shifted above the bias window during t_i and the electron has two possibilities to escape. First, it can tunnel back to the reservoir, which would not contribute to a net current (dashed arrow in ii,iii) or, if the levels are either aligned (iii) or forming a downhill path (ii), electrons can also tunnel through the DQD, which would result in a negative current as we observe in Fig. R2a. When the DQD levels are detuned further, the level of the right QD is not accessible during t_i so only the tunneling back to the reservoir is possible and the net current is zero. In i, the levels are above the bias window during the whole pulse and no electron is entering the DQD, which explains the zero net current. We have added the following sentence to the manuscript:

"The region of negative current in Fig. 4a can be attributed to charge pumping occurring outside of the gate configurations where the measurement scheme presented in Fig. 3a operates."

Figure R2: **a** Charge stability diagram of the DQD with a square pulse ($f_{\text{rep}} = 316$ MHz, $t_p = 165$ ps, $t_i = 3$ ns, $A_p = 228 \mu\text{eV}$) applied to the left FG. **b** Schematics of the level configurations during the initialization (t_i , blue lines) and the shorter manipulation (t_p , red lines) part of the pulse at the positions in gate space i-iv in a.

9. Comparing with Fig. 2d, 2f and Fig. S2, I find that the transition lines locate at different positions in Fig. 2e. Is the figure mislabeled?

We thank the reviewer for her/his very careful investigation. Indeed the figure was mislabeled. We have corrected this error in Fig. 2e in the manuscript as can be seen below.

Figure R3: Revised Figure 2: The ticks in Fig. 2e are now at the correct position. Revised caption: [...] **h** Resonant excitation frequency as a function of δV_L (see panel e). The dashed red curve shows a fit according to $hf = \sqrt{(\alpha \delta V_L)^2 + \Delta^2}$ while the shaded blue bands mark the $\pm 1\sigma$ confidence interval. The dashed black line is a straight line through the origin with the slope of the lever arm $\alpha = 194.5 \mu\text{eV/mV}$. The grey data points in the background are the results for two distinct sets of FG voltages ($V_L = 3.475 \text{ V}$, $V_R = 3.38 \text{ V}$, $\Delta/2h \approx 1.54 \text{ GHz}$ and $V_L = 3.555 \text{ V}$, $V_R = 3.36 \text{ V}$, $\Delta/2h \approx 7.87 \text{ GHz}$). Inset: Schematic representation of the electrochemical potentials in a QD illustrating the process of PAT.

Reviewer 2:

The manuscript "Coherent charge oscillations..." is devoted to the study of driven quantum dots on the base of bilayer graphene. Such structures are perspective given their high tunability and the ability to manipulate electric, spin and valley degrees of freedom. Making use of quantum interferometry, the authors provide the protocol for inducing and detecting coherent charge oscillations and define the charge decoherence times. Thus, the authors demonstrate experimentally the ability of individual control, manipulation, and characterization of the double-quantum-dot devices based on bilayer graphene.

The work is done on the highest experimental level and the paper is very well written.

We thank reviewer for her/his time and effort reading carefully through the manuscript and appreciate the questions and suggestions raised below.

I have only a few comments below. In addition, my report addresses several questions:

- What are the noteworthy results?
The work is devoted to a fine experimental realization of the double quantum dot on the base of a bilayer graphene. This structure is reliably driven, controlled, and measured. Coherent charge oscillations are demonstrated and their analysis allows for obtaining decoherence times, determined from both the Fourier analysis and the photon-assisted tunnelling processes.
- Will the work be of significance to the field and related fields? How does it compare to the established literature?
The work addresses the topical object and uses modern techniques of control, interferometry and spectroscopy. The priority is in observing the multi-photon coherent oscillations in a bilayer graphene double quantum dots for the first time. This seems to be important and promising in the field of quantum engineering and technology.
- Does the work support the conclusions and claims, or is additional evidence needed? Is the methodology sound? Does the work meet the expected standards in your field?
The work is very well written, with all the conclusions and claims well justified. The methodology, including experimental realizations and theoretical analysis, is sound and is on a very high level.
- Is there enough detail provided in the methods for the work to be reproduced?
Experimentally, maybe. Theoretically, yes, there is enough information to reproduce details, including photon-assisted tunnelling (Fig. 2), coherent oscillations (Fig.4), and their Fourier analysis (Fig. 5).

We thank the reviewer for elaborating on the experimental and theoretical details of our study. We checked the Methods section and included additional information on the measurements method to assure the reproducibility of our work. The Methods section now includes the following information on fabrication and measurement techniques:

- The fabrication method of the heterostructure and the thicknesses of the hBN dielectrics.
- Geometries, material composition and fabrication method of the gate structure on top of the heterostructure.
- The important metrics of the home-built PCB.
- The measurement method of the DC current.
- The devices to generate the sine and square pulses used in the LZSM and PAT measurements.

- The RF attenuation at room temperature and at cryogenic temperatures.
- The room temperature rise time of the measurement setup (see Fig. R1).
- The voltages of the other gates, which are kept constant throughout the experiment and the applied magnetic field.

A few particular comments follow.

The work is devoted to a structure based on bilayer graphene. Moreover, it seems that the major ambition of the work is the first demonstration of the coherent oscillations in the quantum structure based on this material. However, I feel that there is not enough information in the paper to convince the reader that coherent control in such structures is so much important and perspective. Very little information is given about the graphene structure in this work and almost no references are given about the *bilayer* graphene, while there are many recent and visible papers on the bilayer graphene, including review articles, e.g. <https://iopscience.iop.org/article/10.1088/0034-4885/76/5/056503> <https://doi.org/10.1016/j.physrep.2016.07.003>

We follow the suggestion of the reviewer and have added references to McCann et al. and Rozhkov et al. in the introduction.

We agree with the reviewer, that charge coherence experiments are a well-established area of study. Charge coherence has received significant attention in conventional semiconductors such as GaAs [1]–[3] and Si [4]. These material systems are well understood by now. However, despite the recent experimental progress in the field of bilayer graphene (BLG) QD devices, no coherent oscillations have been reported, so far.

It is worth highlighting that it is not immediately evident that charge coherence could be observed in van-der-Waals heterostructures of two-dimensional materials – such as the device investigated in our work – as well. There are significant differences between a van-der-Waals heterostructure and a two-dimensional electron gas in a conventional semiconductor. In a heterostructure of BLG encapsulated between two crystals of hexagonal boron nitride (hBN), the electron wavefunction will extend into the hBN/BLG interface. Any type of charge disorder present at that artificially made interface, e.g. charged impurities, can potentially limit the charge coherence time. In contrast, the two-dimensional electron gas in a GaAs/AlGaAs heterostructure is typically fabricated at a depth of 55-90 nm below the surface [5], [6] while in Si/SiGe heterostructures it is typically buried at 30-60 nm below the surface [7], [8] which makes it less susceptible to disorder at the interface to the dielectric.

Furthermore, the thickness of the hBN crystals measures typically about 30 nm, i.e. the gates in closest vicinity to the QDs (back gate and split gates) are only 30 nm away. The small distance together with the large spatial overlap lead to a strong capacitive coupling of these gates to the QDs. This in turn can make the QDs sensitive to gate-induced charge noise limiting the charge decoherence time.

Despite the expected increase in sensitivity to charge noise (whether interface or gate-induced), we were able to demonstrate the first coherent control of the charge degree of freedom in any 2D heterostructure. The determined charge decoherence times of 400-500 ps are en par with those reported in GaAs. As outlined above, this finding is not at all obvious and it is an indicator for a low density of trap states and hence a quality measure for the van-der-Waals interface.

Hence, the novelty of our work lies in the technology, as we measure charge coherence on a fundamentally different material system. The fact that the extracted quantities for the decoherence time are on the same order of magnitude as for conventional semiconductors is an important novel result on the way targeting spin and valley qubits in BLG.

The investigation of charge noise is highly relevant also beyond the application as a potential charge qubit: Charge noise can be a source of decoherence and relaxation in any type of qubit which allows

electrical control. In particular, charge noise can couple to the spin degree of freedom via spin-orbit coupling or mediated by a magnetic field gradient induced by a micromagnet. This type of noise has been identified even as the limiting source of decoherence in a Si spin qubit [9]. In addition, calculations show that charge noise is the dominant cause for the relaxation of valley states in BLG QDs in the regime of low magnetic fields (i.e. at low energy splitting) [10]. Hence, charge decoherence measurements are a suitable tool to investigate BLG QD devices and their suitability for application as spin or valley qubits. We included parts of the discussion presented above into the manuscript. It now reads:

"However, despite these recent experimental advances, no coherent oscillations of either charge, spin or valley states have yet been reported in BLG quantum devices. A priori, it is not obvious that charge coherence can be observed in van der Waals heterostructures such as BLG encapsulated between hexagonal boron nitride (hBN) crystals. In contrast to QDs in semiconductor heterostructures based on GaAs [5], [6] and SiGe [7], [8], which are buried tens of nm below the dielectric interface, the electron wave function of a BLG QD extends into the hBN making it susceptible charge noise present due to disorder at the BLG/hBN interface and to impurities in the hBN. So far, no light has been shed on the role of charge noise in graphene QDs."

"[...] These time scales are en par with those reported for advanced GaAs QDs [2], [3] which is a first indicator for low charge noise and an important quality measure for the van-der-Waals interfaces in the BLG-based heterostructure."

We have added the following sentence to the Discussion section:

"Furthermore, temperature-dependent decoherence measurements allow to quantify electron-phonon coupling, an intrinsic source of decoherence in QDs, as demonstrated in GaAs QDs [11]."

The authors refer to the four papers, by Landau, Zener, Stueckelberg, and Majorana, Refs. [1-4]. (Are these references useful for a reader?) However, in the text, the authors use only LZS and LZ abbreviations, why not LZSM? Note that the original works of LZSM were recently discussed here: <https://doi.org/10.1016/j.physrep.2022.10.002>

We thank the reviewer for pointing out the comprehensive review article by Ivakhnenko et al.. We agree that this work should be cited in the manuscript and have added this reference in the introduction accordingly. However, we regard it as justified to also cite the original work by Landau, Zener, Stueckelberg and Majorana, as well. Thus, we prefer to keep these references, too.

We agree with the reasoning of the reviewer for naming the interference effects Landau-Zener-Stueckelberg-Majorana (LZSM). However, the naming convention is not consistent throughout literature, LZ and LZS are used as well. We have changed the naming to LZSM throughout the manuscript. For example, the first sentence of the abstract now reads:

"The coherent dynamics of a quantum mechanical two-level system passing through an anti-crossing of two energy levels can give rise to Landau-Zener-Stückelberg-Majorana (LZSM) interference."

The caption of Fig. 1(a) says that either the conductive channel or the reservoirs is/are highlighted in blue; but it seems that there is no blue colour in Fig. 1(a).

We thank the reviewer for this helpful comment and agree with her/him that the blue colour is hardly visible. We have changed it to a brighter colour which highlights the conductive channel and the reservoirs. For a direct comparison of the old and the revised Fig. 1a, see Fig. R4 below.

Also, I guess that in addition to V_L , applied to the left finger, there is V_R applied to the right fin-

Figure R4: Revised Figure 1a **a** old Figure, **b** revised Figure.

ger, as we can see in Fig. 1(b). As an optional comment, maybe it is worthwhile to tell something about the right finger, in the caption of Fig. 1(a), after telling about the left one.

We have added a label V_R in Fig. 1a and mention it in the caption. It now reads as:

"Our so-called finger gate left is connected to a bias tee which allow applying DC (V_L) and AC voltages (V_{AC}). The voltage V_R is applied to the right finger gate."

Reviewer 3:

In this manuscript, the authors present an investigation on the microwave spectroscopy of molecular states in a BLG DQDs. A noteworthy contribution of this study is the observation of coherent charge oscillations within the graphene-based system, which has not been reported previously. For the most part, the results and discussions are clear, and the interpretation of the data is warranted. However, there are several concerns regarding this study that authors need to clarify.

We thank the reviewer for his/her careful examination of our manuscript.

This report focuses solely on a single case (or another, in Fig. S7) of inter-dot tunnel coupling. The BLG DQDs examined in this study are defined by gate voltages, which allow for control over the inter-dot coupling by manipulating these voltages. The authors also mention that "This indicates that the inter-dot tunnel coupling can be sensitively tuned by the voltages applied to the finger gates." In light of this result, it would be valuable to examine the outcomes associated with different tunnel couplings.

We agree with the reviewer that an investigation of charge coherence at different tunnel couplings would be an interesting study. We want to point out that coherent oscillations were observed at several combinations of FG voltages in this device (see Fig. S7, Fig. R5). However, it is important to note that the tuning of the FG voltages in this device not only affects the inter-dot coupling but also simultaneously modifies the tunneling rates to the reservoirs. Consequently, when we decrease the tunnel coupling, we must extend the initialization time due to the overall reduced tunneling rates. In Fig. R5, the effect of small tunneling rates can be observed: the decreased tunnel coupling leads to a larger LZS oscillation period, as expected, but at the same time, the signal-to-noise ratio in the average current measurement drops. Likewise, for higher tunnel couplings, matching the oscillation frequency would require increasing the overall pulse frequencies. However, as can be seen in Fig. 4b, we already reach the frequency limits of our measurement setup in the intermediate tunnel coupling regime.

In conclusion, a more detailed investigation of charge coherence at different tunnel couplings would require either full control over all the tunneling rates to the reservoirs and the inter-dot tunnel coupling independently from each other, or the incorporation of a charge detector which allows for measurements of single tunnelling events. We plan to implement these improvements in future devices but unfortunately goes beyond what we can do at the moment. We added Fig. R5 to the supplementary with the following description:

"Fig. S9 depicts LZSM measurements at a different charge occupation where the overall tunneling rates have decreased. This is evident from the reduced current as well as the reduced LZSM oscillation period in detuning (a) and pulse duration t_p (b)."

Figure R5: Additional data set for lower tunnel coupling. **a** Triple point with applied square pulse $t_p = 160$ ps, $t_i = 5$ ns and $V_p = 100$ mV. **b** Line cut along the x-axis in **a** at $V_R = 3.334$ V as function of the pulse duration t_p .

The PAT measurement (see Fig. 1h) would clearly estimate the inter-dot tunnel coupling constants. Pioneering studies of microwave response of DQD in GaAs (T.H. Oosterkamp et al., Nature 395, 873, 1998) have reported the control of the inter-dot tunnel coupling constants by gate voltages.

It is true, that photon-assisted tunneling is a good spectroscopy tool to determine the tunnel coupling. In fact, we performed similar measurements as T.H. Oosterkamp et al. did for five different tunnel couplings in [20]. Fig. R6 shows our measurements of the tunnel coupling in three different regimes by performing PAT spectroscopy at different gate configurations. Since the tunability of the tunnel coupling in similar devices has been already shown before [21], we restricted ourselves on the coherent oscillations of charge in the regime presented in the manuscript. Nevertheless, we gladly follow the suggestion of the reviewer and include the additional PAT measurements in Fig. R6 to the manuscript. The revised Fig. 2 is shown in Fig. R3 (see page 7 of this document).

Figure R6: Tunnel coupling measured by PAT spectroscopy for different regimes of the device. Red: For lower tunnel coupling, the measurement method is limited, due to the leakage currents at zero detuning. This can be seen in the higher error of the value. Blue: Original data set of the main manuscript. Green: The tunnel coupling is strongly tuned by the right finger gate and results in a much higher tunnel coupling.

Furthermore, the period of coherent charge oscillation relies on the inter-dot tunnel coupling constants. Thus, if the authors can present results demonstrating coherent charge oscillation while controlling the tunnel coupling, it would signify the tremendous tunability of quantum dots within the graphene-based system.

We acknowledge the importance of investigating the period of coherent charge oscillation in relation to the inter-dot tunnel coupling constants within our graphene-based quantum dot system. Since we do not have access to parts of the measurement equipment, unfortunately we can not perform the measurements on coherent oscillations depending on the pulse duration t_p . Nevertheless, in response to the reviewers suggestion, we have performed additional experiments of the amplitude dependency of the LZS measurements, specifically targeting the control of inter-dot tunnel coupling. Fig. R7a shows a second device used for LZS measurements while controlling the inter-dot tunnel coupling. Here, additional to the FGs used to form the DQD (blue), a barrier gate in between (red), which is separated by another layer of 15 nm Al_2O_3 , can be used to control the tunnel coupling. To perform the LZS measurements, we use a sine pulse of frequency $f = 10$ GHz and amplitude V_{AC} generated from the Agilent E8257D microwave source and apply it to the left FG. Fig. R7c shows line cuts through the triple point of the charge transition $(1, 0) - (0, 1)$ (shown in Fig. R7b) as a function of detuning for three different barrier gate voltages ($V_b = -6.7$ V, -6.9 V and -7.5 V). By fitting the dependency of the current on the detuning for resonant tunneling ($\Delta/2 \gg \Gamma_{out}, \Gamma_{in}$)

$$I(\varepsilon) = \frac{4et_c^2/\Gamma_{out}}{1 + (2\varepsilon/h\Gamma_{out})^2}, \quad (1)$$

to the line cuts in Fig. R7c, the tunnel coupling $\Delta = 2t_c$, as well as the tunnel rates to the leads, Γ_{out} , can be estimated for the three regimes. The results for Δ show a strong trend on the barrier gate, as already has been seen in earlier measurements [21]. The same trend can be observed in the LZS measurements in Fig. R7d, where not only the overall current increases for higher inter-dot tunnel coupling, but also the shape of the LZS pattern changes. By using the model for transport during the LZS measurement from Floquet scattering theory in the limit $hf \gg \Delta$ [16]

$$I(\varepsilon, A) = \frac{e}{\hbar} \frac{\Gamma_{in}\Gamma_{out}}{2(\Gamma_{in} + \Gamma_{out})} \sum_{n=-\infty}^{\infty} \frac{\Delta_n}{(\varepsilon - nhf)^2 + \Delta_n^2 + (\Gamma_{in} + \Gamma_{out})^2/4}, \quad (2)$$

with $\Delta_n = J_n(A/hf)\Delta$, we calculate the LZS pattern for the parameters found by the fit of eq. 1 in Fig. R7c. The results depicted in Fig. R7e, reproduce the overall current and the qualitative LZS pattern which we observe for the three regimes of different tunnel coupling in Fig. R7d.

Figure R7: LZS measurements with a second sample for three different tunnel couplings. **a** False color scanning electron microscope (SEM) picture of the gate structure of the second device. The DQD is formed below the two FGs (blue) by applying the voltages V_L and V_R . Additionally, a sine pulse can be applied to the left FG (V_{AC}). In between the two FGs, a barrier gate is used to control the inter-dot tunnel coupling. **b** Triple point of the charge transition $(1, 0) - (0, 1)$ with a bias voltage of $V_{SD} = 1$ mV and $V_b = -7.4$ V. **c** Line cut through the triple point in b as function of the amplitude of the sine pulse with a frequency of $f = 10$ GHz and barrier voltages of $V_b = -6.7$ V, -6.9 V and -7.5 V. **d** Line cut of the measurements above, where the tunnel coupling $\Delta/2$ could be estimated by a Lorentzian fit of Eq. 1. **e** Numeric calculations of the LZS pattern in d with the parameters found in c.

The current manuscript could be read as saying that coherent charge oscillations have been ob-

served in BLG QDs that accidentally possess a moderate inter-dot tunneling coupling. While this outcome is undoubtedly novel, it may not meet the criteria for publication in Nature Communications, which emphasizes research with a broad impact. The authors would need to show that they were able to control BLG DQD with tunnel coupling and coherent charge oscillations tunably. Therefore, I would recommend that the authors consider submitting their current manuscript to a more specialized journal, such as Phys. Rev. Lett.

We appreciate the reviewer's suggestion publishing our work in Phys. Rev. Lett. We regret that the current manuscript seems to be perceived as an observation of '*coherent charge oscillations [...] that accidentally possess a moderate inter-dot coupling.*'. However, we indeed measured the tunnel coupling in three distinct regimes, as shown in Fig. R6. Subsequently, we carefully chose the regime for the coherent charge oscillations for our study. The decision to omit the additional data from the manuscript was made to maintain a clear presentation of our main findings. To improve on the point criticised by the reviewer, we included the additional PAT measurements at three different gate configurations to the manuscript to show the tunability of the tunnel coupling in this device and to clarify that the results are not achieved by accident. As already mentioned above, the measurement method used in this work is limited by both the signal-to-noise ratio for low tunnel coupling and the sampling rate of the AWG for high tunnel coupling. Therefore, we are not able to provide additional studies of the time-dependent measurements for the distinct regimes. Fig. 2 in the manuscript includes now the additional photon-assisted tunneling data and can be seen in Fig. R3. To make it clearer, that the intermediate tunnel coupling was not achieved accidentally, we added the following paragraph to the manuscript:

"Additionally, similar measurements were conducted for a set of two different FG voltages ($V_L = 3.475$ V, $V_R = 3.38$ V, and $V_L = 3.555$ V, $V_R = 3.36$ V), which yield $\Delta/2h \approx 1.54$ GHz and $\Delta/2h \approx 7.87$ GHz, respectively. For the following measurements the regime of intermediate tunnel coupling (2.1 GHz) was chosen."

Other specific comments to be addressed are as follows.

1. In terms of the formation of a quantum two-level system, it would be valuable to elucidate the energy difference of the single-particle level for each individual dot. This information is essential for a comprehensive understanding of the studied system. Notably, the presence of OD-excited states seems to be observed in the lower triple points of Fig. 1(c), and inside the triangle of Fig. S1.

We thank the reviewer for this suggestion. We have to point out that the data presented in this work are measured in the 'few electron regime' as written in the manuscript. The excited states visible in Fig. 1c and Fig. S1 indeed are part of the energy spectrum. When energetically accessible and contributing to the transport during the pulsed measurements, the excited states would either show up in the photon-assisted tunneling spectroscopy as additional resonances [14] or they would also be visible in the LZS measurements (see Fig. S6,S7) [16]. The fact that this is not the case in our measurements can be explained by the high valley g-factor ($g_v \approx 15-30$) in BLG and the relatively high perpendicular magnetic field ($B_{\perp} = 1.8$ T) which is applied, so that the system is already valley polarized ($\Delta E_v \geq 1.5$ meV). Also the spin Zeeman effect shifts the states with opposite spin about $\Delta E_s \approx 210$ μ eV in energy, which is beyond the energy scale, where we are operating the system (PAT, Fig. 2: $\varepsilon \leq 140$ μ eV, LZS, Fig. 4: $\varepsilon \leq 200$ μ eV, Fig. 5: $\varepsilon \leq 300$ μ eV). To better clarify the influence of the magnetic field and the excited state spectrum, we added the following paragraph to the manuscript:

"In the PAT measurements, also excited states could play a role, if energetically accessible [14]. The absence of these excited states can be explained by the out-of-plane mag-

netic field of 1.8 T applied to the device, which polarizes the valley states (valley splitting of ≈ 1.5 meV) and also partly the spin states (spin splitting of ≈ 210 μ eV)."

2. Concerning the PAT experiment, it would be advantageous to demonstrate the microwave power dependence (PAT pattern) and provide an evaluation indicating that the peak heights exhibit behavior consistent with the square of the Bessel function. This evaluation would enhance the understanding of the obtained results.

We thank the Reviewer for suggesting to include these measurements to our work. We indeed performed these at a different gate configuration. Fig. S3a shows a line cut along the V_L axis through the triple point at $V_R = 3.3601$ V and a bias voltage of $V_{SD} = 5$ μ V as function of the applied microwave power at $f = 23$ GHz. We observe the appearance of higher order PAT resonances when increasing the power, reaching up to three resonances at the maximum applied power of 25 dBm. Fig. S3b shows the current averaged in a small window around one resonance ($n=1,2,3$) as function of the applied power on a linear scale (V_{rms}). The amplitude of the PAT resonance follows the same behavior as the squared Bessel function depicted in the Inset and shows the expected proportionality on the parameter $a = e\beta V_{rms}/hf$. Due to the power limitation of 25 dBm of the signal generator, only up to one oscillation can be seen in the amplitude dependence. For the evaluation of Δ , the system was operated in the low power regime, where only the first order PAT appears. Following the reviewers suggestion, we included the Figure to our supplementary information. The added paragraph to Section II reads:

"Higher order PAT resonances that emerge as the microwave power is increased are shown in Fig. S3a. Here, a line cut as function of finger gate voltage, V_L , is shown as a function of applied power. The elevated power level renormalizes the tunnel coupling by the squared Bessel function and can be described by the function $f = \sqrt{(\alpha\delta V_L)^2 + J_0(a)^2\Delta^2}/h$, where $a = e\beta V_{rms}/hf$. In the case of low applied power ($a \ll 1$), $J_0^2 \approx 1$, the tunnel coupling can be estimated directly by fitting Eq. 1 in the main text [22]. Fig. S3b displays the averaged current across the PAT resonances of orders $n = 1, 2, 3$. A comparison with the inset, presenting the squared Bessel functions, reveals the anticipated proportionality of higher order PAT [22]. The power for the evaluation of Δ is set to a minimum, where only the first order PAT process is visible."

Figure R8: **a** Line cut through a triple point at $V_R = 3.3601$ V, $V_{SD} = 5$ μ V as function of the power of the applied sine pulse with a frequency of $f = 23$ GHz. **b** Averaged current through the DQD system along the PAT peaks of the first, second and third order ($n=1,2,3$) as function of the effective power V_{rms} . Inset: Squared Bessel function of the first to third order as function of the parameter $a = e\beta V_{\text{rms}}/hf$.

References

- [1] G. Cao, H.-O. Li, T. Tu, L. Wang, C. Zhou, M. Xiao, G.-C. Guo, H.-W. Jiang, and G.-P. Guo, “Ultrafast universal quantum control of a quantum-dot charge qubit using Landau–Zener–Stückelberg interference,” *Nat. Commun.*, vol. 4, p. 1401, Jan. 2013, ISSN: 2041-1723. DOI: 10.1038/ncomms2412.
- [2] J. R. Petta, A. C. Johnson, C. M. Marcus, M. P. Hanson, and A. C. Gossard, “Manipulation of a Single Charge in a Double Quantum Dot,” *Phys. Rev. Lett.*, vol. 93, no. 18, p. 186802, Oct. 2004, ISSN: 1079-7114. DOI: 10.1103/PhysRevLett.93.186802.
- [3] K. D. Petersson, J. R. Petta, H. Lu, and A. C. Gossard, “Quantum Coherence in a One-Electron Semiconductor Charge Qubit,” *Phys. Rev. Lett.*, vol. 105, no. 24, p. 246804, Dec. 2010, ISSN: 1079-7114. DOI: 10.1103/PhysRevLett.105.246804.
- [4] M. F. Gonzalez-Zalba, S. N. Shevchenko, S. Barraud, J. R. Johansson, A. J. Ferguson, F. Nori, and A. C. Betz, “Gate-Sensing Coherent Charge Oscillations in a Silicon Field-Effect Transistor,” *Nano Lett.*, vol. 16, no. 3, pp. 1614–1619, Mar. 2016, ISSN: 1530-6984. DOI: 10.1021/acs.nanolett.5b04356.
- [5] F. Martins, F. K. Malinowski, P. D. Nissen, E. Barnes, S. Fallahi, G. C. Gardner, M. J. Manfra, C. M. Marcus, and F. Kuemmeth, “Noise Suppression Using Symmetric Exchange Gates in Spin Qubits,” *Phys. Rev. Lett.*, vol. 116, no. 11, p. 116801, Mar. 2016, ISSN: 1079-7114. DOI: 10.1103/PhysRevLett.116.116801.
- [6] T. Botzem, R. P. G. McNeil, J.-M. Mol, D. Schuh, D. Bougeard, and H. Bluhm, “Quadrupolar and anisotropy effects on dephasing in two-electron spin qubits in GaAs,” *Nat. Commun.*, vol. 7, no. 11170, pp. 1–5, Apr. 2016, ISSN: 2041-1723. DOI: 10.1038/ncomms11170.
- [7] S. G. J. Philips, M. T. Mądzik, S. V. Amitonov, S. L. de Snoo, M. Russ, N. Kalhor, C. Volk, W. I. L. Lawrie, D. Brousse, L. Tryputen, B. P. Wuetz, A. Sammak, M. Veldhorst, G. Scappucci, and L. M. K. Vandersypen, “Universal control of a six-qubit quantum processor in silicon,” *Nature*, vol. 609, pp. 919–924, Sep. 2022, ISSN: 1476-4687. DOI: 10.1038/s41586-022-05117-x.
- [8] D. R. Ward, D. Kim, D. E. Savage, M. G. Lagally, R. H. Foote, M. Friesen, S. N. Coppersmith, and M. A. Eriksson, “State-conditional coherent charge qubit oscillations in a Si/SiGe quadruple quantum dot,” *npj Quantum Inf.*, vol. 2, no. 16032, pp. 1–6, Oct. 2016, ISSN: 2056-6387. DOI: 10.1038/npjqi.2016.32.
- [9] J. Yoneda, K. Takeda, T. Otsuka, T. Nakajima, M. R. Delbecq, G. Allison, T. Honda, T. Kodera, S. Oda, Y. Hoshi, N. Usami, K. M. Itoh, and S. Tarucha, “A quantum-dot spin qubit with coherence limited by charge noise and fidelity higher than 99.9%,” *Nat. Nanotechnol.*, vol. 13, pp. 102–106, Feb. 2018, ISSN: 1748-3395. DOI: 10.1038/s41565-017-0014-x.
- [10] L. Wang and G. Burkard, *Private communication*,
- [11] T. Hayashi, T. Fujisawa, H. D. Cheong, Y. H. Jeong, and Y. Hirayama, “Coherent Manipulation of Electronic States in a Double Quantum Dot,” *Phys. Rev. Lett.*, vol. 91, no. 22, p. 226804, Nov. 2003, ISSN: 1079-7114. DOI: 10.1103/PhysRevLett.91.226804.
- [12] F.-M. Jing, Z.-Z. Zhang, G.-Q. Qin, G. Luo, G. Cao, H.-O. Li, X.-X. Song, and G.-P. Guo, “Gate-Controlled Quantum Dots Based on 2D Materials,” *Adv. Quantum Technol.*, vol. 5, no. 6, p. 2100162, Jun. 2022, ISSN: 2511-9044. DOI: 10.1002/qute.202100162.
- [13] J. Boddison-Chouinard, A. Bogan, N. Fong, K. Watanabe, T. Taniguchi, S. Studenikin, A. Sachrajda, M. Korkusinski, A. Altintas, M. Bieniek, P. Hawrylak, A. Luican-Mayer, and L. Gau-dreau, “Gate-controlled quantum dots in monolayer WSe₂,” *Appl. Phys. Lett.*, vol. 119, no. 13, Sep. 2021, ISSN: 0003-6951. DOI: 10.1063/5.0062838.

- [14] C. J. van Diepen, P. T. Eendebak, B. T. Buijtenorp, U. Mukhopadhyay, T. Fujita, C. Reichl, W. Wegscheider, and L. M. K. Vandersypen, "Automated tuning of inter-dot tunnel coupling in double quantum dots," *Appl. Phys. Lett.*, vol. 113, no. 3, p. 033101, Jul. 2018, ISSN: 0003-6951. DOI: 10.1063/1.5031034.
- [15] K. Wang, C. Payette, Y. Dovzhenko, P. W. Deelman, and J. R. Petta, "Charge Relaxation in a Single-Electron Si/SiGe Double Quantum Dot," *Phys. Rev. Lett.*, vol. 111, no. 4, p. 046801, Jul. 2013, ISSN: 1079-7114. DOI: 10.1103/PhysRevLett.111.046801.
- [16] F. Forster, G. Petersen, S. Manus, P. Hänggi, D. Schuh, W. Wegscheider, S. Kohler, and S. Ludwig, "Characterization of Qubit Dephasing by Landau-Zener-Stückelberg-Majorana Interferometry," *Phys. Rev. Lett.*, vol. 112, no. 11, p. 116803, Mar. 2014, ISSN: 1079-7114. DOI: 10.1103/PhysRevLett.112.116803.
- [17] L. Banszerus, S. Möller, C. Steiner, E. Icking, S. Trellenkamp, F. Lentz, K. Watanabe, T. Taniguchi, C. Volk, and C. Stampfer, "Spin-valley coupling in single-electron bilayer graphene quantum dots," *Nat. Commun.*, vol. 12, p. 5250, Sep. 2021, ISSN: 2041-1723. DOI: 10.1038/s41467-021-25498-3.
- [18] A. Kurzman, Y. Kleeorin, C. Tong, R. Garreis, A. Knothe, M. Eich, C. Mittag, C. Gold, F. K. de Vries, K. Watanabe, T. Taniguchi, V. Fal'ko, Y. Meir, T. Ihn, and K. Ensslin, "Kondo effect and spin-orbit coupling in graphene quantum dots - Nature Communications," *Nat. Commun.*, vol. 12, no. 6004, pp. 1–6, Oct. 2021, ISSN: 2041-1723. DOI: 10.1038/s41467-021-26149-3.
- [19] T. Ota, K. Hitachi, and K. Muraki, "Landau-Zener-Stückelberg interference in coherent charge oscillations of a one-electron double quantum dot," *Sci. Rep.*, vol. 8, no. 5491, pp. 1–8, Apr. 2018, ISSN: 2045-2322. DOI: 10.1038/s41598-018-23468-2.
- [20] T. H. Oosterkamp, T. Fujisawa, W. G. van der Wiel, K. Ishibashi, R. V. Hijman, S. Tarucha, and L. P. Kouwenhoven, "Microwave spectroscopy of a quantum-dot molecule," *Nature*, vol. 395, pp. 873–876, Oct. 1998, ISSN: 1476-4687. DOI: 10.1038/27617.
- [21] L. Banszerus, A. Rothstein, E. Icking, S. Möller, K. Watanabe, T. Taniguchi, C. Stampfer, and C. Volk, "Tunable interdot coupling in few-electron bilayer graphene double quantum dots," *Appl. Phys. Lett.*, vol. 118, no. 10, Mar. 2021, ISSN: 0003-6951. DOI: 10.1063/5.0035300.
- [22] W. G. van der Wiel, S. De Franceschi, J. M. Elzerman, T. Fujisawa, S. Tarucha, and L. P. Kouwenhoven, "Electron transport through double quantum dots," *Rev. Mod. Phys.*, vol. 75, no. 1, pp. 1–22, Dec. 2002, ISSN: 1539-0756. DOI: 10.1103/RevModPhys.75.1.

REVIEWERS' COMMENTS

Reviewer #1 (Remarks to the Author):

The authors have addressed my comments well and amended the manuscript accordingly. I am ready to support its publication in Nature Communications with one minor point left:

I agree with the authors that graphene-based van-der-Waals heterostructure is a new material platform, compared with conventional semiconductors. The extended introduction explains their differences well. Meanwhile, I think discussions on the possibilities and differences of charge coherence experiments in other 2D materials are also important, since this work is considered as the first coherent control of charge degree of freedom in the diverse 2D material family. This will provide a complete landscape of the field and broaden the impact of this work, especially given the fact that similar high-quality double dot devices have already been demonstrated in MoS₂ system (Sci. Adv. 3, e1701699 (2017)). Therefore, I urge the authors to include discussions on other 2D materials into the manuscript as well.

Reviewer #2 (Remarks to the Author):

The authors have taken into account very accurately all the comments of the three referees. The authors presented very detailed and convincing responses, where they responded to the referees and described what and how was introduced in the text, and how much additional work they did, starting from studying new references and ending with doing additional measurements. I recommend the publication of the manuscript in its present form.

Reviewer #3 (Remarks to the Author):

I consider that all my comments are replied adequately. The authors have made the explanation of the experimental works much more clear than the previous version. I recommend publication in Nature Communications.

I think it would be better for the reader to include the results and discussions of Fig. R7 in the Supplementary Information of the paper. This is because the results from the another device with the similar device structure confirm the reproducibility of the graphene quantum dot device.

However, the journal will also open the referee reports, so the authors/editors can decide which is better.

Reviewer 1:

The authors have addressed my comments well and amended the manuscript accordingly. I am ready to support its publication in Nature Communications with one minor point left: I agree with the authors that graphene-based van-der-Waals heterostructure is a new material platform, compared with conventional semiconductors. The extended introduction explains their differences well. Meanwhile, I think discussions on the possibilities and differences of charge coherence experiments in other 2D materials are also important, since this work is considered as the first coherent control of charge degree of freedom in the diverse 2D material family. This will provide a complete landscape of the field and broaden the impact of this work, especially given the fact that similar high-quality double dot devices have already been demonstrated in MoS₂ system (Sci. Adv. 3, e1701699 (2017)). Therefore, I urge the authors to include discussions on other 2D materials into the manuscript as well.

We thank reviewer for her/his positive opinion on our revised manuscript and the recommendation for publication. We thank for the suggestion to extend the discussion on charge coherence experiments in other 2D materials.

We have extended the introduction as follows:

More recently, *2D materials, such as bilayer graphene (BLG) and transition metal dichalcogenides* have emerged as potentially interesting alternative *materials* with appealing properties for highly controllable QDs, interesting for hosting spin and valley qubits. [...]

In addition, QDs have also been realised in WSe₂ and MoS₂ monolayers [1], [2], which are of interest due to their substantial intrinsic spin-orbit coupling and potential for light-matter coupling. However, despite these recent experimental advances, no coherent oscillations of either charge, spin or valley states have yet been reported in quantum devices based on 2D materials.

We have extended the discussion section by the following sentences:

Charge coherence experiments can also be extended to DQDs in other 2D materials such as WSe₂ and MoSe₂ [1]. The high sensitivity of this type of experiment can be exploited to characterise material-dependent properties such as defect-induced and interfacial charge noise and electron-phonon coupling, which are expected to be different from BLG.

Reviewer 2:

The authors have taken into account very accurately all the comments of the three referees. The authors presented very detailed and convincing responses, where they responded to the referees and described what and how was introduced in the text, and how much additional work they did, starting from studying new references and ending with doing additional measurements. I recommend the publication of the manuscript in its present form.

We thank reviewer for her/his very positive opinion on our revised manuscript and the recommendation for publication.

Reviewer 3:

I consider that all my comments are replied adequately. The authors have made the explanation of the experimental works much more clear than the previous version. I recommend publication in Nature Communications.

I think it would be better for the reader to include the results and discussions of Fig. R7 in the Supplementary Information of the paper. This is because the results from the another device with the similar device structure confirm the reproducibility of the graphene quantum dot device.

However, the journal will also open the referee reports, so the authors/editors can decide which is better.

We thank reviewer for her/his positive opinion on our revised manuscript and the recommendation for publication. We agree with the reviewer that results from another device confirm the reproducibility. Thus, we have included Fig. S10 in the Supplementary Information, which shows the device, a charge stability diagram and a LZSM dataset.

References

- [1] Z.-Z. Zhang, X.-X. Song, G. Luo, G.-W. Deng, V. Mosallanejad, T. Taniguchi, K. Watanabe, H.-O. Li, G. Cao, G.-C. Guo, F. Nori, and G.-P. Guo, "Electrotunable artificial molecules based on van der Waals heterostructures," *Sci. Adv.*, vol. 3, no. 10, Oct. 2017, ISSN: 2375-2548. DOI: 10.1126/sciadv.1701699.
- [2] J. Boddison-Chouinard, A. Bogan, N. Fong, K. Watanabe, T. Taniguchi, S. Studenikin, A. Sachrajda, M. Korkusinski, A. Altintas, M. Bieniek, P. Hawrylak, A. Luican-Mayer, and L. Gaudreau, "Gate-controlled quantum dots in monolayer WSe₂," *Appl. Phys. Lett.*, vol. 119, no. 13, Sep. 2021, ISSN: 0003-6951. DOI: 10.1063/5.0062838.